# Nonlinear Behaviour of Critical Points for a Simple Neural Network

**G. Welper**                                                                     *gerrit.welper@ucf.edu*
*Department of Mathematics*
*University of Central Florida*
*Orlando, FL 32816, USA*

**Reviewed on OpenReview:** *https://openreview.net/forum?id=wfdG2PEOHS*

## Abstract

In severely over-parametrized regimes, neural network optimization can be analyzed by linearization techniques as the neural tangent kernel, which shows gradient descent convergence to zero training error, and landscape analysis, which shows that all local minima are global minima. Practical networks are often much less over-parametrized, and training behaviour becomes more nuanced and nonlinear. This paper contains a fine grained analysis of the nonlinearity for a simple shallow network in one dimension. We show that the networks have unfavourable critical points, which can be mitigated by sufficiently high local resolution. Given this resolution, all critical points satisfy $L_2$ loss bounds of optimal adaptive approximation in Sobolev and Besov spaces on convex and concave subdomains of the target function. These bounds cannot be matched by linear approximation methods and show nonlinear and global behaviour of the critical point's inner weights.

## 1 Introduction

In this paper, we analyze nonlinear aspects of neural network training for a simple model problem in supervised learning: For samples $x_i$ and data $y_i = f(x_i)$ generated by some unknown target function $f$, find a neural network $f_\theta$ with weights $\theta$ by minimizing the least squares loss. To motivate the results, we first review some common approaches in the literature.

**Landscape Analysis** Gradient descent can easily get stuck in local minima. That this fact does not harm neural network training is the purview of landscape analysis. It aims to demonstrate that either the loss has no local minima, in favour of saddle points, or all local minima have small loss value and therefore provide good trained networks. Indeed, the papers Soudry & Carmon (2016); Kawaguchi (2016); Nguyen & Hein (2017); Ge et al. (2018); Du & Lee (2018); Soltanolkotabi et al. (2019); Venturi et al. (2019); Kawaguchi et al. (2019); Kawaguchi & Huang (2019) show that local minima are global minima, either under strong assumptions, or over-parametrization with more network width than number of samples. Absent such assumptions, one needs to be more careful, e.g. the papers Swirszcz et al. (2017); Safran & Shamir (2018); He et al. (2020); Ding et al. (2022); Jentzen & Riekert (2024), find local minima that are not global. A more fine grained analysis of the landscape is included in He et al. (2020), which finds valleys of path connected local minima.

Since these results are mixed, matching local and global minima may be to strong a goal and on may be content with a simpler question:

(Q1) Do critical points have favorable properties and what are these?

To address this question, first note that ultimately we are not interested in a good training error, but rather in a good generalization error $\inf_\theta \|f_\theta - f\|^2_{L_2(\mathcal{P})}$ for some probability measure $\mathcal{P}$ that generates the input

samples $x_i$. In general, it is difficult to understand the exact nature of the global optimum, but it is much more feasible to understand upper bounds of the form

$$\inf_\theta \|f_\theta - f\| \lesssim n(\theta)^{-r}, \quad f \in K, \tag{1}$$

where $n(\theta)$ is an indicator for the network size, like width, depth or total number of weights and $r > 0$ an asymptotic rate. Similar to the no-free-lunch theorem, such bounds cannot work for arbitrary $f$, which is why we restrict them to some compact set $K$. Typically, it bounds Sobolev, Besov or Barron norms or other smoothness properties of the permissible targets $f$. Inequalities of type (1) are common in approximation theory and have been studied extensively for neural networks. A literature overview is given later in the introduction.

We use this perspective to ease the characterization of local minima. If they do not match the global minimum, can they match their scaling behaviour

$$\|f_\theta - f\| \lesssim n(\theta)^{-r}, \quad f \in K, \quad \theta \text{ is a critical point of the training loss?} \tag{2}$$

Such results are well established for partial differential equations (PDEs), where $f_\theta$ is a nonlinear approximation method like adaptive finite elements or wavelets and $f$ the solution of a PDE Cohen et al. (2002); Morin et al. (2002); Binev et al. (2004). Similar results also exits for shallow neural networks, when trained with greedy algorithms Siegel & Xu (2022b); Siegel et al. (2023) instead of gradient descent.

**Linearization Arguments** In over-parametrized regimes, typically with more network width than training samples, gradient descent training does not move the network weights far from their random initialization. As a result, one can obtain accurate descriptions of the training dynamics by linearising the network at the initial value. Careful analysis then provides exponential gradient descent convergence to zero training loss. A common representative of this approach is the neural tangent kernel (NTK) introduced in Jacot et al. (2018); Li & Liang (2018); Allen-Zhu et al. (2019); Du et al. (2019b;a), and refined in Zou et al. (2020); Arora et al. (2019a;b); Su & Yang (2019); Lee et al. (2019); Song & Yang (2019); Zou & Gu (2019); Kawaguchi & Huang (2019); Chizat et al. (2019); Oymak & Soltanolkotabi (2020); Ji & Telgarsky (2020); Nguyen & Mondelli (2020); Bai & Lee (2020); Cao & Gu (2020); Chen et al. (2021); Song et al. (2021); Lee et al. (2022); Gentile & Welper (2022); Welper (2024b;a); Keene & Welper (2024).

Contrary to this analysis, much of the promise of neural networks relies on their severe non-linearity, leading to e.g. high expressivity and excellent function approximation properties, even in high dimensions. Can these be exploited by gradient descent training? If we consider less over-parametrization, or even slightly under-parametrized regimes, the weights can move farther from their initial and break the linear dominance in the training dynamics. Empirical studies Vyas et al. (2023) (see also Lee et al. (2020); Seleznova & Kutyniok (2022)) on image classification datasets show that in such regimes networks perform better than extremely wide networks with dominantly linear behaviour. A theoretical understanding of these regimes is still largely unknown. This leads to a second question:

(Q2) Can training in under-parametrized or only slightly over-parametrized regimes exploit the nonlinear nature of neural networks?

To this end, it is instructive to look at classical approximation methods, where $f_\theta$ is replaced by e.g. splines, finite elements or wavelets. These depend nonlinearly on $\theta$ if adaptivity is used and linearly if not. The nonlinear variations strictly include the linear ones so that $\inf_\theta \|f_\theta^{nonlinear} - f\| \le \inf_\theta \|f_\theta^{linear} - f\|$. Nonetheless, in the upper error bounds (1) this does neither change the number of degrees of freedom (weights) $n(\theta)$ nor the (maximal) rate $r$. It does change, however, the size of the compact sets $K^{linear} \subset K^{nonlinear}$ for which the given rates can be achieved, with the latter being significantly larger.

In summary, if we want to establish approximation results (2) for neural network critical points with nonlinear compact sets $K^{nonlinear}$, we have to carefully exploit the nonlinear nature of the networks and can no longer rely on vanilla NTK analysis.

**New Contributions**   In this paper, we address questions (Q1) and (Q2) for the very simple model problem

$$f_\theta(x) := \sum_{r=1}^{m} w_r \sigma(x - b_r), \tag{3}$$

with ReLU activation, in a one dimensional interval $x \in D \subset \mathbb{R}$, trained on the $L_2$ loss $\|f_\theta - f\|_{L_2(D)}$. This is probably the simplest choice with nonlinear weight dependence (of the $b_r$), non-convex loss and fully understood approximation behaviour both in linear ($b_r$ untrained) and non-linear ($b_r$ trained) cases. The continuous loss simplifies the analysis and places the problem in an under-parametrized regime, independent of the width $m$. Empirical losses, with large numbers of samples, are expected to show similar behaviour by classical arguments in statistics and machine learning, different from the over-parametrized regime, where their application is more complicated.

Although this setup may seem simple, it contains two challenges:

1. The problem does have bad local minima.

2. Large compact sets $K^{nonlinear}$ in the approximation bounds (2) cannot be achieved by linear approximation methods and require careful global placement of the nonlinear inner weights $b_r$.

The first result shows this global placement for all critical points of the loss function in the infinite width limit: If we order the inner weights $b_0 \leq \cdots \leq b_m$ the normalized grid size satisfies

$$\lim_{\substack{m \to \infty \\ x \in [b_{r-1}, b_r]}} m(b_r - b_{r-1}) = \text{constant}|f''(x)^{-2/5}|, \tag{4}$$

with possibly a different constant for each interval on which $f$ is strictly convex or concave. The factor $m \sim 1/h$ is reciprocal to the uniform grid size $h$ and used for normalization. The right hand side shows that the breakpoints $b_r$ are close wherever the second derivative $f''(x)$, and hence the local approximation difficulty, is large. Generally, this requires global movement of breakpoints $b_r$ dependent on $f$, from initial locations independent of $f$. For finite $m$, analogous arguments show that at critical points of the loss the breakpoints equidistribute the local smoothness

$$\|f''\|_{L_{2/5}([b_{r-1}, b_r])} = \text{constant},$$

again on intervals $D_{\mathcal{I}}$ where $f$ is convex or concave. With standard approximation theory, this leads to approximation errors of the type

$$\|f_\theta - f\|_{L_2(D_{\mathcal{I}})} \lesssim |\mathcal{I}|^{-2}\|f''\|_{L_{2/5}(D_{\mathcal{I}})},$$

where $|\mathcal{I}|$ is the number of breakpoints in the respective intervals. To avoid bad local minima, these results require the critical points to have sufficient local resolution $|b_r - b_{r-1}|$ so that $f$ does not have highly oscillatory features between breakpoints that are imperceptible to the gradient. The rigorous statements are in Theorems 3.3 and 3.4 and an example for the conditions is given in Section 5.

The results demonstrate approximation errors (2) on subdomains where $f$ is convex or concave with $K := K_{2/5} := \{f \in L_2(D) : \|f''\|_{L_{2/5}(D)} \leq 1\}$. A subtle, but crucial, observation is that $f''$ is measured in the very weak $L_{2/5}$ (quasi-) norm (or Besov spaces in Section B.3), which allows us to achieve high approximation orders for fairly rough functions $f$. These are not possible for purely linear approximation methods (by Kolmogorov $n$-width lower bounds) and therefore demonstrate that finding local critical points of the loss landscape allows us to exploit some nonlinearity of the neural networks.

**Infinite Width Limit**   Mean field theory of neural networks Chizat & Bach (2018); Mei et al. (2018); Rotskoff & Vanden-Eijnden (2018); Sirignano & Spiliopoulos (2020) takes the infinite width limit

$$\frac{1}{m}\sum_{r=1}^{m} w_r \sigma(v_r^T x) \quad \to \quad \int w\sigma(v^T x)\, d\mu(v, w)$$

for some limiting measure $\mu$ and then analyzes training of the infinite networks. For comparison, the limits of the gird size (4) are taken in different order: We first compute the gradient, decouple the computation of $b_r$ from $w_r$ and then take the limit afterwards.

**Beyond Linearization**   Some recent papers analyze neural network training beyond the NTK regime. For example, Damian et al. (2022); Lee et al. (2024) demonstrate results for two layer networks that cannot be achieved by kernel methods for polynomials $g(Ux)$ that depend only on a few dimension by the inner matrix $U \in \mathbb{R}^{r \times d}$ with $r \ll d$.

**Approximation**   Universal approximation theorems Cybenko (1989); Hornik et al. (1989); Barron (1993); Zhou (2020); Lu et al. (2017); Hanin & Sellke (2017) show that neural networks can approximate any function arbitrarily well. Since this is true for virtually all approximation methods in practical use, it is important to quantify the approximation error more closely. This usually leads to errors bounds of type (1), which are studied extensively for neural networks. If the compact set $K$ consists of functions with bounded Sobolev or Besov smoothness, results can be found in Gribonval et al. (2022); Gühring et al. (2020); Opschoor et al. (2020); Li et al. (2019); Suzuki (2019), or for improved rates that surpass classical methods for the price of discontinuous weight assignments in Yarotsky (2017; 2018); Yarotsky & Zhevnerchuk (2020); Daubechies et al. (2022); Shen et al. (2019); Lu et al. (2021). Compact sets $K$ specifically tailored to neural networks include Barron and related spaces Bach (2017); Klusowski & Barron (2018); Weinan et al. (2022); Li et al. (2020); Siegel & Xu (2020; 2022a); Bresler & Nagaraj (2020); Parhi & Nowak (2021); Unser (2023). Overviews are in Pinkus (1999); DeVore et al. (2021); Weinan et al. (2020); Berner et al. (2022).

In the majority of neural network approximation results, the weights are hand-picked and only few papers show approximation properties of gradient descent trained neural networks Jentzen & Riekert (2022); Ibragimov et al. (2022); Drews & Kohler (2022); Kohler & Krzyzak (2022); Gentile & Welper (2022); Welper (2024b;a). These heavily rely on the outermost linear layer, or a NTK linearization and therefore show approximation guarantees only for compact sets $K$ that can be well approximated by linear methods. Larger, nonlinear classes $K$ can, to best of our knowledge, so far only be proven for greedy training algorithms Siegel & Xu (2022b); Siegel et al. (2023) which rely on another non-convex optimization problem in each step.

**Notations**   We use $c$ for generic constants that can be different in each occurrence, but are independent of $f$ and the network width $m$. We abbreviate $a \leq cb$, $a \geq b$ and $ca \leq b \leq cb$ by $a \lesssim b$, $a \gtrsim b$ and $a \sim b$, respectively. We define $[m] := \{1, \ldots, m\}$ and $\mathbb{P}^r$ as all polynomials of degree at most $r$. We use Sobolev $\|\cdot\|_{W^{s,p}(D)}$ and Besov $\|\cdot\|_{B_q^s(L_p(D))}$ norms with their usual definitions, stated in Section B.1. For any interval $I$, we denote the corresponding $L_2$ inner product by $\langle \cdot, \cdot \rangle_I$.

## 2   Approximation By Piecewise Linear Functions

Before we state the main results of the paper, we review relevant approximation properties of the neural networks. The set of all networks of type 3

$$\Upsilon_m := \left\{ f_\theta(\cdot) = \sum_{r=1}^m w_r \sigma(\cdot - b_r) \,\middle|\, w_r, b_r \in \mathbb{R} \right\}.$$

corresponds exactly to *continuous piecewise linear (CPwL)* functions in one dimension

$$\Sigma_m := \{f \mid f \text{ is continuous piecewise linear with } m \text{ breakpoints}\},$$

often referred to as first order free knot splines or finite elements. To discuss the benefits of nonlinearity, we compare them with the simpler linear class

$$\Sigma_m^u := \{f \in \Sigma_m \mid \text{uniform distance between neighbouring breakpoints}\},$$

corresponding to networks with untrained inner biases $b_r$ and hence convex loss. Notice that the latter set $\Sigma_m^u$ is linear, while the former $\Upsilon_m = \Sigma_m$ is nonlinear and hence we refer to them as *linear* and *nonlinear approximation methods*. Their approximation errors are precisely understood:

$$\inf_{\phi \in \Sigma_m^u} \|\phi - f\|_{L_2} \lesssim C m^{-2} |f|_{B_2^2(L_2)},$$
$$\inf_{\phi \in \Sigma_m} \|\phi - f\|_{L_2} \lesssim C m^{-2} |f|_{B_{2/5}^2(L_{2/5})}. \tag{5}$$

These correspond exactly to the approximation bound in the introduction if we define $K_{s,p} := \{f \in L_2(D) : |f|_{B^2_p(L_p)} \le 1\}$ with $p = 2$ and $p = 2/5$.

Up to minor differences, the Besov norms $|f|_{B^s_p(L_p)} \approx \|f^{(s)}\|_{L_p}$ are equivalent to Sobolev norms, which bound the $s$-th derivative of $f$ in $L_p$. The former are technical, but usually preferred in approximation theory because they are well behaved for $p < 1$, as used in the nonlinear bound above. For orientation, these spaces are often arranged as in Figure 1. The sets $K_{s,p}$ become larger with decreasing $s$ and decreasing $p$. By Sobolev embedding theorems, one may trade some $p$ for some $s$ so that all spaces $(s,p)$ above the dashed line in the figure are contained in $L_2$ and thus $K_{s,p} \subset K_{0,2}$. See Section B.1 for definitions and DeVore & Lorentz (1993); DeVore (1998) for more details.

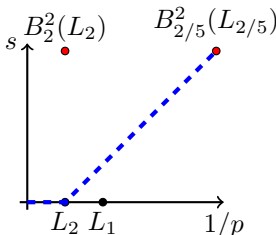

Figure 1: Diagram of Besov spaces. Each point $(1/p, s)$ corresponds to one space $B^s_p(L_p)$ for $s > 0$ and $L_p$ for $s = 0$.

Let us compare the linear approximation $\Sigma^u_m$ with the nonlinear approximation $\Sigma_m$. First observe that in (5) the rate $m^{-2}$ is identical for both methods. Generally, piecewise linear approximation does not achieve higher rates, even if $f$ admits more smoothness. However, since $\|\cdot\|_{L_{2/5}(D)} \lesssim \|\cdot\|_{L_2(D)}$, the smoothness conditions for nonlinear approximation are much weaker. For example $f_\epsilon = \text{sigmoid}(x/\epsilon)$ has norms

$$\|f''_\epsilon\|_{L_p(D)} \sim \epsilon^{\frac{1}{p}-2}, \qquad \|f''_\epsilon\|_{L_2(D)} \sim \epsilon^{-\frac{3}{2}}, \qquad \|f''_\epsilon\|_{L_{2/5}(D)} \sim \epsilon^{\frac{1}{2}}. \qquad (6)$$

Indeed, the second derivative is of size $\epsilon^{-2}$ in a region $[-c\epsilon, c\epsilon]$ and negligible outside. Thus $\|f''_\epsilon\|_{L_p(D)} \approx \epsilon^{-2}\|1\|_{L_p([-c\epsilon, c\epsilon])} \sim \epsilon^{-2}\epsilon^{\frac{1}{p}}$. As $\epsilon$ goes to zero and $f_\epsilon$ converges to a jump function, the $L_2$ norm blows up, whereas the $L_{2/5}$ norm remains bounded. This provides significantly better approximation bounds in (5) for nonlinear approximation. If we use Besov spaces instead of the second derivative, this extends to the jump function itself, which can be approximated by nonlinear methods up to order $m^{-2}$, whereas linear approximation only achieves order $< m^{-1/2}$.

While the linear approximation has a fixed number of breakpoints $b_r$ near the jump or sharp gradients of $f_\epsilon$, the adaptive approximation can allocate more resources where $f$ is complicated. Indeed, algorithms and proofs for the approximation bounds (5), aim for breakpoints that equidistribute the local errors

$$\|f_\theta - f\|_{L_2([b_r, b_{r-1}])} = \text{constant for all } r$$

or closely related the local smoothness

$$\|f''\|_{L_{2/5}([b_r, b_{r-1}])} = \text{constant for all } r. \qquad (7)$$

Finally note that the bounds (5) are sharp in several ways. For example the best possible rate linear approximation methods can achieve for functions $f$ in the class $K_{2,p} \subset K_{2,2/5}$ with $2 < p \le 1$ is $m^{-2+\frac{1}{p}-\frac{1}{2}} < m^{-2}$, see Lorentz et al. (1996), Chapter 14, Theorem 1.1. Therefore, if we can find critical points of the neural network (9) loss that achieves second order $m^{-2}$ error on the class $K^{2/5}$, it must exploit the nonlinearity of the network.

## 3 Main Result

**Setup** Maybe the simplest neural network with non-convex training objective and fully understood approximation properties is

$$F_{W,V,B}(x) := B_0 + W_0 x + \sum_{r=1}^{m} W_r \sigma(V_r x - B_r) \tag{8}$$

in one dimension, which generates the piecewise linear functions $\Sigma_m$ of the last section. By the property $\sigma(ax) = a\sigma(x)$ for $a \geq 0$ of ReLU activations, the separation of $W_r$ and $V_r$ is redundant and we simplify the network to

$$f_{w,b} := b_0 + w_0 x + \sum_{r=1}^{m} w_r \sigma(x - b_r),$$

with appropriately redefined weights. The case $V_r < 0$ requires a slight technicality and is considered in the following lemma. The lemma also shows that critical points of the loss carry over to the simplified network

**Lemma 3.1.** *Let $D$ be an interval. Assume $F_{W,V,B}$ is a critical point of the loss $\min_{W,V,B} \|F_{W,V,B} - f\|^2_{L_2(D)}$. Then $F_{W,V,B}$ is piecewise linear with $\bar{m} \leq m$ breakpoints*

$$b_r = B_r / V_r, \qquad\qquad\qquad \text{for all } r \text{ with } W_r, V_r \neq 0.$$

*Furthermore, there is a network $f_{w,b}(x) = F_{W,V,B}(x)$ of width $\bar{m}$ with the same breakpoints that is a critical point of the loss $\min_{w,b} \|f_{w,b} - f\|^2_{L_2(D)}$.*

The proof is in Section B.4. Since our main theorems characterize critical points, it suffices to consider the simplified variant of the networks. Finally, the term $b_0 + w_0 x$ can be generated by two ReLUs $\sigma(x - b_r)$, one with breakpoint at the left boundary and one more left of the domain. Therefore, we drop $b_0$ and $w_0$ and arrive at the network

$$f_\theta(x) := \sum_{r=1}^{m} w_r \sigma(x - b_r), \tag{9}$$

that we consider throughout the paper.

We train the network with loss

$$\ell(\theta) := \frac{1}{2} \|f_\theta - f\|^2_{L_2(D)} \tag{10}$$

on some finite domain $D \subset \mathbb{R}$ and some target function $f \in L_2(D)$. This matches the infinite sample limit of the least squares loss and places us in an under-parametrized regime similar to classical statistics. Although the ReLU activation has kinks, this loss is strongly differentiable for all weights $\theta$. Indeed, it suffices to consider the network as a map $\theta \to f_\theta(\cdot)$ from parameters to $L_2(D)$ functions. This topology is sufficiently weak to render the map differentiable, unlike the regular pointwise topology, which does not. See Gentile & Welper (2022) for details.

**Cleanup** Gradient descent and related methods converge to critical points,

$$\nabla \ell(\theta) = 0, \tag{11}$$

which we examine more closely in the following. To ease the theoretical analysis, we start with some notational cleanup, which does not alter the actual network. First, we drop inactive neurons with $w_r = 0$. Second, we join neurons with identical bias $b_r$ into one neuron and adjust the outer weights $w_r$ accordingly. Third, we drop neurons for biases $b_r$ outside of the domain $D$, except for the largest $b_r$ left of the domain, which influences the left boundary value of $f_\theta$. Finally, we add one artificial breakpoint $b_{\bar{m}}$ at the right end of $D$, which does not change $f_\theta$ inside $D$, but avoids technicalities. This yields $\bar{m} \leq m$ neurons, which we reorder according to

$$b_0 < \cdots < b_{\bar{m}} \tag{12}$$

and denote as *cleaned critical breakpoints*. These define intervals $I_r := [b_{r-1}, b_r]$ of length $h_r := |I_r|$. We denote two consecutive intervals by $I_{r+} := I_r \cup I_{r+1}$ and $h_{r+} := |I_{r+}|$.

**Equidistribution** We have seen in the last section that optimal asymptotic approximation rates are achieved by equidistributing local errors or smoothness via careful placement of the breakpoints $b_r$. It is instructive to start with an informal discussion of the infinite width limit. To this end, we define the *grid size limit*

$$h(x) := \lim_{\substack{m \to \infty \\ x \in I_r}} m h_r,$$

where for every $m$ we choose the interval $I_r$ with width $h_r$ that contains $x$. For a uniform grid, we have grid size $h_r = |D|/m$ and therefore $h(x) = mh_r = |D|$. For non-uniform grids, $h(x)$ measures the ratio $h_r/h$ between the actual local grid size $h_r$ and the uniform grid size $h = |D|/m$, up to a global factor. If the limit exists, $h(x)$ is given by the following lemma.

**Lemma 3.2.** *Let $f$ be smooth and for every $m$ let $b_r$, $r \in [\bar{m}]$ be cleaned critical breakpoints* (11), (12). *If the limit $h(x)$ exists, it satisfies*

$$h(x) = c_I f''(x)^{-2/5},$$

*with possibly a different constant $c_I$ on each interval $I$ for which $f''(x)$ is non-zero.*

For finite $m$, a careful perturbation analysis leads to the first main theorem.

**Theorem 3.3.** *Let $\theta$ be a critical point* (11), *with cleaned breakpoints in ascending order 12. For $r, s \in \{2, \ldots, \bar{m}\}$, let $\mathcal{I} = \{r, r+1, \ldots, s\}$ be a set of consecutive neurons with $D_{\mathcal{I}} := \bigcup_{k \in \mathcal{I}} I_k$ and*

$$\max \left\{ h_{k+}^{\frac{1}{2} - \frac{1}{p}} \|f^{(3)}\|_{L_p(I_{k+})}, \ h_{k+}^{1 - \frac{1}{q}} \|f^{(4)}\|_{L_q(I_{k+})}, \ \right\} \le C \min_{x \in I_{k+}} |f''(x)| \tag{13}$$

*for some $1 < q, p \le \infty$ and some sufficiently small constant $C > 0$ independent of $f$ and $h_k$. Then for $l, k \in D_{\mathcal{I}}$ we have equidistribution*

$$\|f''\|_{L_{2/5}(I_l)} \sim \|f''\|_{L_{2/5}(I_k)}.$$

This is precisely the equidistribution (7) used in the proofs of CPwL approximation bounds (5). We discuss the result and the major assumption (13) after the approximation theorem below. In short, high oscillations strictly contained in one interval $I_r$ are imperceptible to the gradient and therefore can lead to bad critical points. The assumption ensures that we have enough breakpoints to fully resolve such oscillations.

**Approximation** Once we have established equidistribution, approximation results can be obtained along standard lines.

**Theorem 3.4.** *Let $\theta$ be a critical point* (11), *with cleaned breakpoints in ascending order 12. For $r, s \in \{2, \ldots, \bar{m}\}$, let $\mathcal{I} = \{r, r+1, \ldots, s\}$ be a set of consecutive neurons with $D_{\mathcal{I}} := \bigcup_{k \in \mathcal{I}} I_k$ and*

$$\max \left\{ h_{k+}^{\frac{1}{2} - \frac{1}{p}} \|f^{(3)}\|_{L_p(I_{k+})}, \ h_{k+}^{1 - \frac{1}{q}} \|f^{(4)}\|_{L_q(I_{k+})}, \ \right\} \le C \min_{x \in I_{k+}} |f''(x)| \tag{14}$$

*for some $1 < q, p \le \infty$ and some sufficiently small constant $C > 0$ independent of $f$ and $h_k$. Then*

$$\|f_\theta - f\|_{L_2(D_{\mathcal{I}})} \lesssim |\mathcal{I}|^{-2} \|f''\|_{L_{2/5}(D_{\mathcal{I}})}. \tag{15}$$

**Discussion** *Approximation Result:* Note that $|\mathcal{I}|$ is the number of breakpoints in $D_{\mathcal{I}}$ and therefore is analogous to the number of breakpoints $\bar{m}$ on the full domain. Therefore, any critical point subject to the given conditions achieves asymptotically optimal CPwL approximation on subdomains $D_{\mathcal{I}} \subset D$ and can properly utilize the nonlinearity of the network. If we have multiple subdomains $D_{\mathcal{I}_1}$ and $D_{\mathcal{I}_2}$, the total allocation of breakpoints on each of these may be suboptimal.

*Resolution:* The main purpose of assumption 13 is to prevent bad critical points: Oscillations of the target $f$ that are strictly contained inside one interval $I_r$ do not change the gradient and can cause unfavorable critical points. The assumption is satisfied if the network's local resolution $h_r$ is sufficiently high to capture all fine grained features of $f$. For comparison classical adaptive CPwL approximation methods contain "data oscillation" terms in their error bounds. See Section 5 for a more careful discussion.

*Smoothness:* The fourth order smoothness in (13) is higher than the smoothness in the approximation bounds and seems to contradict the discussion in Section 2. Indeed, it does rule out limiting cases like jump functions. This is also the reason why we can use Sobolev type norms instead of Besov norms. However, functions with large gradients as in example (6) are permissible. In this case assumption (13) provides some a-posteriori bounds on the resolution necessary to achieve equidistribution and adaptive approximation errors. However, even if this requires large networks none of the constants enters the approximation error itself: As soon as the networks are big enough, we obtain nonlinear approximation bounds with small constants only dependent on the favourable $\|f''\|_{L_{2/5}}$ smoothness bound.

*Convex/Concave:* Assumption (13) implicitly entails that $f$ is concave or convex on each subdomain $D_{\mathcal{I}}$. While it is not clear if this is strictly necessary, longer stretches with $f''(x) = 0$ seem problematic: In these $f$ is linear and can be approximated with zero local error. Then any breakpoint in this region has no good gradient information for its placement.

*Left Boundary:* The networks are zero $f_\theta(x) = 0$ for all $x < b_0$ left of the leftmost breakpoint. To avoid dealing with this boundary condition, we exclude the corresponding interval $I_1$, by requiring $s, r \geq 2$ in the definition of $\mathcal{I}$.

*Comparison with NTK Theory:* In over-parametrized NTK regimes the weights do not move far from their initialization during gradient descent training. This implies that uniformly initialized breakpoints $b_r$ remain uniform and can generally not equidistribute local errors. Accordingly, the approximation error achieved after training is bounded by $\|f_\theta - f\|_{L_2(D)} \lesssim m^{-\alpha}\|f\|_{B_2^s(L_2(D))}$ in Gentile & Welper (2022) for some constants $\alpha \geq 0$ and $0 \leq s \leq 1/2$. In particular, the smoothness is measured in the $L_2$ norm as for uniform CPwL approximation and not in a suitable larger $L_p$ norm as for adaptive CPwL approximation.

*Inner Weights:* The equidistribution results carry over to more standard networks $F_{W,V,B}$, with inner weights $V_r$, as defined in (8). Indeed, by Lemma B.2, the breakpoints of a critical point $F_{W,V,B}$ match the breakpoints of a critical point $f_{w,b}$, for which Theorem (3.3) provides equidistribution on convex/concave sub-domains.

## 4 Proof Idea

This section contains a short overview over the proof. The optimization of the outer weights $w_r$ is convex and therefore fairly simple. On the other hand, the optimization of the inner weights $b_r$ is non-convex and the main objective of the prove is to demonstrate their equidistribution in Theorem 3.3. Once this property is established, the approximation Theorem 3.4 follows by standard arguments DeVore (1998). The proof proceeds in several steps.

1. *Critical Points:* Define the spaces

$$X := \text{span}\{\partial_{w_r} f_\theta \mid r \in [m]\} = \text{span}\{\sigma(x - b_r) \mid r \in [m]\},$$
$$\dot{X} := \text{span}\{\partial_{b_r} f_\theta \mid r \in [m]\} = \text{span}\{w_r \dot{\sigma}(x - b_r) \mid r \in [m]\}$$

of the partial derivatives and the residual $\kappa := f_\theta - f$. Then, by taking linear combinations, it is easy to see that the critical point conditions

$$\partial_{\theta_r} \ell(\theta) = \langle \kappa, \partial_{\theta_r} f_\theta \rangle = 0,$$

are equivalent to

$$\langle \kappa, v \rangle = 0, \qquad\qquad v \in X + \dot{X}. \qquad (16)$$

2. *Eliminate $w_r$:* In the critical point conditions the residual $\kappa$ depends on both $w_r$ and $b_r$. To show equidistribution, which depends on $b_r$ only, we eliminate $w_r$ from the equations. To this end, define the $L_2$-orthogonal complement space $X^\perp$ so that

$$X + \dot{X} = X \oplus X^\perp, \qquad\qquad X \perp X^\perp.$$

Since $X$ is the span of all neurons $\sigma(x - b_r)$, we have $f_\theta \in X$ and therefore $\langle \kappa, v \rangle = \langle f_\theta - f, v \rangle = \langle f, v \rangle$. for all $v \in X^\perp$ by orthogonality. Thus, the critical point condition implies

$$\langle f, v \rangle = 0, \quad v \in X^\perp,$$

This condition does not depend on $w_r$ any longer and guarantees equidistribution, as we show in the following steps of the proof.

3. *Characterization of the Complement space $X^\perp$:* We construct basis functions

$$\varphi_r(x) = \begin{cases} h_r \acute{\phi}''(x), & x \in I_r \\ -h_{r+1} \grave{\phi}''(x), & x \in I_{r+1} \\ 0, & \text{else} \end{cases}$$

for $X^\perp$ supported on two consecutive intervals $I_r$ and $I_{r+1}$. The critical point condition then yields $\langle f, \varphi \rangle = 0$ and integration by parts

$$h_r \langle f'', \acute{\phi}_r \rangle_{I_r} = h_{r+1} \langle f'', \grave{\phi}_{r+1} \rangle_{I_{r+1}}. \tag{17}$$

Since the functions $\acute{\phi}$ and $\grave{\phi}$ are non-negative bump functions, the smoothness conditions of the main theorems imply that

$$\|f\|_{L_{1/2}(I_r)} \sim h_r \langle f'', \acute{\phi}_r \rangle_{I_r} = h_{r+1} \langle f'', \grave{\phi}_{r+1} \rangle_{I_{r+1}} \sim \|f\|_{L_{1/2}(I_{r+1})}. \tag{18}$$

4. *Refined Analysis:* While the last two equations provide equidistribution on two neighbouring intervals, they are insufficient: (18) has the wrong norm, $L_{1/2}$ instead of $L_{2/5}$, and is too inaccurate when chaining over large numbers of intervals. (17) cannot be chained directly because the functions $\acute{\phi} \neq \grave{\phi}$ are asymmetric. A more refined analysis of the asymmetry and passing to the limit $m \to \infty$ shows that the grid size limit $h(x) = \lim_{m \to \infty} m h_r$ for $x \in I_r$ satisfies the differential equation

$$[h^2 f'']' = \frac{1}{5} h^2 f''',$$

where the left hand side originates from (17) and the right hand side from the asymmetry. This is a first order linear differential equation for $h^2$. Solving it with an integrating factor leads to $[h^2 (f'')^{4/5}]' = 0$ and the extra power $4/5$ leads to proper grid size limit for $L_{2/5}$ equidistribution. The main result Theorem 3.3 follows from a perturbation analysis of this ODE for finite $h_r$.

## 5 Unfavourable Critical Points

### 5.1 Critical Points with High Oscillations

Recall from (16) that critical points are given by the condition

$$\langle f_\theta - f, v \rangle = 0, \qquad\qquad v \in X + \dot{X}.$$

To construct unfavourable critical points, we merely need a perturbation $\varphi$ that is orthogonal to $X + \dot{X}$. Then $f_\theta$ is also a critical point for $f + \varphi$:

$$\langle f_\theta - (f + \varphi), v \rangle = 0, \qquad\qquad v \in X + \dot{X}.$$

It is easy to construct $\varphi$ so that $f_\theta$ is a bad approximation. To provide a simple example, let $f = 0$ so that $f_\theta = 0$ must also be zero. Now choose two neighbouring breakpoints $b_r$ and $b_{r+1}$ and define an oscillation $\varphi$ supported inside $I_r = [b_{r-1}, b_r]$, with some margin to the boundary and orthogonal to all linear functions $\mathbb{P}^1$. Then, we have $\varphi \perp X + \dot{X}$ and $f_\theta = 0$ is a critical point for approximating $0 + c\varphi$ with arbitrarily large approximation error $c\|\varphi\|$. On the other hand, the network $f_\theta$ may have an arbitrary number of breakpoints outside of $I_r$. With optimal placement, they can all be used to approximate $\varphi$ and make the approximation

error arbitrarily small. This simple example is sufficient for our purposes, but the literature contains many more Swirszcz et al. (2017); Safran & Shamir (2018); He et al. (2020); Ding et al. (2022); Jentzen & Riekert (2024).

Note that by construction any small perturbation of the outer weights $w_r$ or the breakpoints $b_r$ does not change the loss for approximating $0 + c\varphi$ so that the large error is imperceptible to gradient based optimizers. In the main theorems, assumption (13) ensures that the network has sufficient resolution so that the target $f$ cannot have any severe sub-grid oscillations. For comparison, classical adaptive CPwL approximation algorithms use error indicators to steer the approximation towards error equidistribution. Theoretical results then include extra "data oscillation" error terms to capture sub-grid oscillations Cohen et al. (2002); Binev et al. (2004); Morin et al. (2002).

## 5.2 Avoiding Critical Points with High Oscillations

This section provides an informal motivation how assumption (13) rules out bad local minima with sub-grid oscillations. To this end, first note that in case the target $f$ is a simple function, like e.g. a second degree polynomial, it clearly cannot oscillate as $\varphi$ in the counter example. Of course this is too strong an assumption, but what if $f$ is close to a second order polynomial? Since second order approximation converges faster than first order approximation, for sufficiently small intervals $I_r = [b_{r-1}, b_r]$, we can expect

$$\inf_{p_2 \in \mathbb{P}^2} \|f - p_2\|_{L_2(I_r)} \le C \inf_{p_1 \in \mathbb{P}^1} \|f - p_1\|_{L_2(I_r)} \le C\|f - f_\theta\|_{L_2(I_r)} \tag{19}$$

for some arbitrarily small constant $C$, with the last implied by the linearity of $f_\theta$ on $I_r$. Denoting the minimizer on the left by $p_{2,r}$, with the triangle inequality, it is easy to show (see e.g. (36)) that

$$(1 + C)^{-1}\|p_{2,r} - f_\theta\|_{L_2(I_r)} \le \|f - f_\theta\|_{L_2(I_r)}.(1 - C)^{-1}\|p_{2,r} - f_\theta\|_{L_2(I_r)}, \tag{20}$$

i.e. the error of approximating $f$ and of approximating the simple function $p_{2,r}$ is about the same. Hence, if we can prove error equidistribution for locally simple functions $p_{2,r}$, we also obtain error equidistribution for $f$

$$\|f - f_\theta\|_{L_2(I_r)} \sim \|p_{2,r} - f_\theta\|_{L_2(I_r)} \sim \|p_{2,s} - f_\theta\|_{L_2(I_s)} \sim \|f - f_\theta\|_{L_2(I_s)}$$

for any two intervals $r$ and $s$. This is a slightly stronger alternative to the equidistribution of smoothness in Theorem 3.3. In summary, given (19), we can reduce equidistribution from arbitrary $f$ to simpler equidistribution of piecewise second degree polynomials.

Before we discuss (19) more carefully, let us apply our observations to the counter example from the last section. If the oscillation $\varphi$ from the example is orthogonal to $\mathbb{P}^1$ and $\mathbb{P}^2$, the best approximations and the network from the example satisfy $p_1 = p_2 = f_\theta = 0$ and we have

$$\|p_{2,r} - f_\theta\|_{L_2(I_r)} = 0, \qquad \qquad \|f - f_\theta\|_{L_2(I_s)} = \|\varphi\|_{L_2(I_s)} \ne 0,$$

which contradicts the norm equivalence (20). Hence, if we can establish (19), severe sub-grid oscillations as in the counter example are not possible.

Let us now consider the condition (19) more carefully. Informally, it follows directly from assumption (13) of the main theorems. Indeed, standard direct approximation inequalities (41) provide the first degree approximation error

$$\inf_{p_1 \in \mathbb{P}^1} \|f - p_1\|_{L_2(I_r)} \lesssim h_r^2 \|f^{(2)}\|_{L_2(I)},$$

and the second degree approximation error

$$\inf_{p_2 \in \mathbb{P}^2} \|f - p_2\|_{L_2(I_r)} \le c h_r^3 \|f^{(3)}\|_{L_2(I)}.$$

Notice that the second degree approximation has a higher power on $h_r$ and therefore is much smaller for small intervals. Assumption (13) contains a concrete criterion when this happens (replacing $h_{r+}$ with $h_r$ for

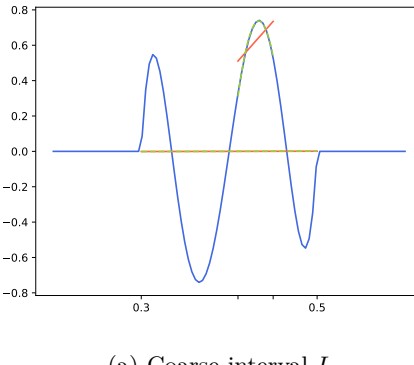

(a) Coarse interval $I_r$

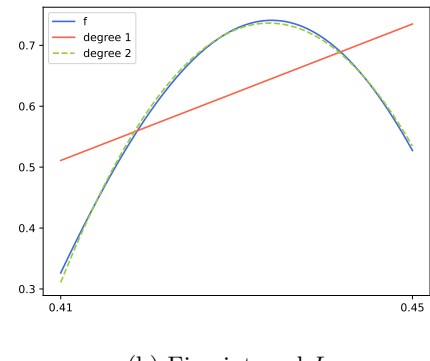

(b) Fine interval $I_r$

Figure 2: Approximation of a oscillatory $f = \varphi = L_5 - L_3$ constructed from Legendre basis polynomials $L_i$. On the coarse interval (left), the first and second degree approximation do not capture the function correctly. On the zoom in to a fine interval (right), the second degree approximation captures the bulk of the function, whereas the first degree approximation does not.

simplicity). It implies

$$
\inf_{p_2 \in \mathbb{P}^2} \|f - p_2\|_{L_2(I_r)} \leq c h_r^3 \|f^{(3)}\|_{L_2(I)} \overset{(13)}{\leq} cC h_r^{\frac{5}{2}} \min_{x \in I_r} |f^{(2)}(x)|
$$

$$
= cC h_r^{\frac{5}{2}} \left( \frac{1}{h_r} \int_{I_r} \min_{x \in I_r} |f^{(2)}(x)|^2 \right)^{\frac{1}{2}} = cC h_r^2 \|f^{(2)}\|_{L_2(I)},
$$

for some constant $C$ that is assumed to be sufficiently small. Direct approximation results tend to be sharp asymptotically, in which case we can directly compare the first and second order approximation to obtain (19):

$$
\inf_{p_2 \in \mathbb{P}^2} \|f - p_2\|_{L_2(I_r)} \lesssim C h_r^2 \|f^{(2)}\|_{L_2(I)} \lesssim C \inf_{p_1 \in \mathbb{P}^1} \|f - p_1\|_{L_2(I_r)}.
$$

A rigorous argument is e.g. in Lemma A.27, applied to $f''$ and constant polynomials $p_2''$. The lemma also includes replacements $\langle f_\theta - f, v \rangle_{I_r} \approx \langle f_\theta - p_{2,r}, v \rangle_{I_r}$ of $f$ by local polynomials for the critical point conditions (16), which require a little more care because of missing absolute values and potential cancellation.

A simple illustration is given in Figure 2. We can see that the approximation on a coarse interval does not capture the oscillatory $f$ correctly. On the fine interval, the second degree approximation is fairly accurate, and captures the bulk of $f$, as required by (20).

## 5.3   Feasibility of Assumptions

When is the main assumption (13) satisfied? A simple guarantee can be given if

- $f$ is smooth $\|f^{(4)}\|_{L_p(D)}, \|f^{(4)}\|_{L_p(D)} \leq \mathcal{S}$ for some $\mathcal{S} > 0$ and $p$ from (13).

- $f$ is uniformly convex (or concave) $f''(x) \geq \mathcal{C}$ for some constant $\mathcal{C} > 0$ and all $x \in D$ (or a convex region as in the main theorems).

In this case, with the crude bound $\|f^{(s)}\|_{L_p(I_r)} \leq \|f^{(s)}\|_{L_p(D)}$, $s = 3, 4$, the assumption reduces to

$$
h_{k+}^{\frac{1}{2} - \frac{1}{p}} \mathcal{S} \leq C\mathcal{C}
$$

for some sufficiently small $C$. This is clearly satisfied if the breakpoints are close so that $h_r = |b_{r-1} - b_r|$ is small. Note that practically the constants $\mathcal{S}$ and $\mathcal{C}^{-1}$ may be large so that we need many breakpoints to

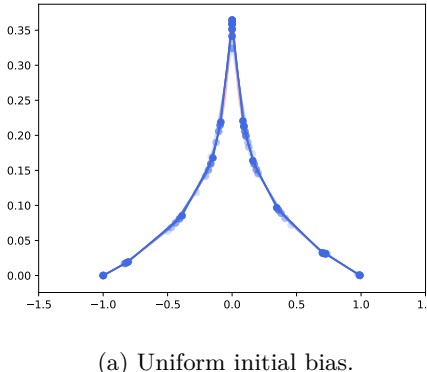

(a) Uniform initial bias.

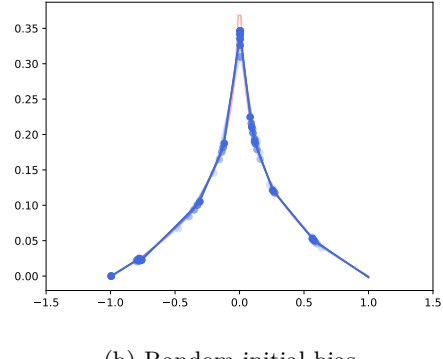

(b) Random initial bias.

Figure 3: Cusp function (21) (red) and neural network approximation (blue) at each 20000-th epoch (light to dark blue). The circle markers show the location of the breakpoints.

apply Theorems 3.3 and 3.4. However, once we can, the resulting equidistribution and approximation errors have standard constants that are independent of $\mathcal{C}$ and $\mathcal{S}$.

We have argued that the condition (13) is simple to verify for a given critical point. A more difficult question is to what extent we can guarantee that these favourable critical points are the limit of gradient descent. We can choose $h_r$ arbitrarily small at the initial breakpoints. By stability of gradient descent, one may argue that two breakpoints that are close at the initial may remain so during training. This would entail (13) also for the gradient descent limit. However, a rigorous argument requires a full understanding of the gradient descent dynamics, which is not considered in this paper and left for future research.

## 6  Numerical Experiments

This section contains some preliminary numerical experiments. We train the target function

$$f(x) = 1 - |x|^{0.1}, \qquad\qquad x \in [-1, 1], \tag{21}$$

which has a cusp at the origin. This entails that $f''$ is not contained in $L_2([-1,1])$, but is contained in $L_{2/5}([-1,1])$ so that we expect a better performance of nonlinear approximation. Intuitively, this is achieved by moving more breakpoints to the vicinity of the cusp. Figure 3 shows the evolution of the networks and breakpoints during gradient descent training with the following setup

- Network architecture (9).
- Network width: 16.
- Learning rate: 0.01.
- 1280 samples in 10 batches.
- 100000 epochs, with a plot of the network every 20000 epochs.

The results show that gradient descent moves the breakpoints towards the cusp at the origin, as expected. However, the convergence is very slow and requires a substantial number of epochs in relation to the simplicity of the problem.

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

# A  Proof of the Main Results

We follow roughly the steps in the proof overview in Section 4:

1. Section A.1 characterizes the critical points, constructs the complement space $X^\perp$ and provides some of its properties.

2. Section A.2 proves the distribution of the grid size limit for infinite width in Lemma 3.2.

3. Section A.3 proves equidistribution in Theorem 3.3.

4. Section A.4 proves the approximation properties of the critical points in Theorem 3.4.

## A.1  Critical Points and the Complement Space $X^\perp$

### A.1.1  Critical Points

We have already seen in the proof overview that the weights $\theta$ are a critical point if and only if $\langle \kappa, v \rangle = 0$ for all $v \in X + \dot{X}$, with residual $\kappa := f_\theta - f$. For reference, this is stated again in the following lemma, together with a characterization of the spaces $X$ and $\dot{X}$.

**Lemma A.1.** *Let $\theta$ be a critical point* (11)*, with cleaned breakpoints in ascending order 12. Then*

$$\langle \kappa, v \rangle = 0, \quad v \in X = \{v \in L_2(D) | v(b_0) = 0, \, v|_{I_r} \in \mathbb{P}^1, \, r = 1, \ldots, \bar{m}\},$$

$$\langle \kappa, v \rangle = 0, \quad v \in \dot{X} = \{v \in L_2(D) | v|_{I_r} \in \mathbb{P}^0, \, r = 1, \ldots, \bar{m}\},$$

*Proof.* In (16), we have seen that the critical points are given by the condition $\langle \kappa, v \rangle = 0$ for all $v$ in the span $X$ of the partial derivatives $\partial_{w_r} \ell$ and in the span $\dot{X}$ of the partial derivatives $\partial_{b_r} \ell$. Hence, it suffices to show that the span of the derivatives matches the two sets in the lemma. Indeed, one readily computes

$$\partial_{w_r} f_\theta = \langle \kappa, \sigma(\cdot - b_r) \rangle, \qquad\qquad \partial_{b_r} f_\theta = \langle \kappa, w_r \dot{\sigma}(\cdot - b_r) \rangle.$$

Clearly ReLU activations $\sigma(\cdot - b_r)$ span piecewise linear functions and their derivatives $w_r \dot{\sigma}(\cdot - b_r)$ span piecewise constants because in the cleanup before Theorem 3.4 we have already dropped all neurons with $w_r = 0$. Finally, by the same cleanup, the leftmost breakpoint maybe inside or outside of $D$, but anyways, we must have $v(b_0) = 0$ because no partial derivative has support left of this point. $\qquad\square$

Note that the network itself is continuous piecewise linear (CPwL) so that $f_\theta \in X$ and the first critical point condition

$$\langle \kappa, v \rangle = 0, \quad v \in X$$

is merely a best $L_2$ projection for the outer weights $w_r$. Together with the second condition

$$\langle \kappa, v \rangle = 0, \quad v \in X + \dot{X} \tag{22}$$

this formally matches a best $L_2$ projection onto discontinuous piecewise linear (DPwL) functions. However, $f_\theta$ is not discontinuous and instead we have to move the breakpoints $b_r$ to satisfy all conditions.

Recall from the proof overview that we split $X + \dot{X}$ into $X$ and the $L_2$-orthogonal complement $X^\perp$:

$$X + \dot{X} = X \oplus X^\perp, \qquad\qquad X \perp X^\perp.$$

Since the neural network $f_\theta$ is contained in $X$, this implies $\langle f, v \rangle = \langle f_\theta - f, v \rangle = \langle \kappa, v \rangle$ for all $v \in X^\perp$ and therefore at a critical point $\langle \kappa, v \rangle = 0$ we have

$$\langle f, v \rangle = 0, \quad v \in X^\perp. \tag{23}$$

Unlike the residual $\kappa = f_\theta - f$, the target $f$ does not depend on the outer weights $w_r$ and therefore the last condition decouples the computation of the inner weights $b_r$ from the outer weights $w_r$. This will be crucial to prove equidistribution, which also depends on the inner weights $b_r$, only.

### A.1.2   The Complement Space $X^\perp$

In this section, we construct an explicit basis for the complement space $X^\perp$. As is customary for e.g. finite elements, we first define suitable functions on the reference interval $\hat{I} := [0,1]$ and then use the bijective affine linear transform

$$T_r \colon \hat{I} \to I_r, \qquad T_r(\hat{x}) = (b_r - b_{r-1})\hat{x} + b_{r-1}, \qquad T_r' = h_r, \qquad (T_r^{-1})' = h_r^{-1}$$

to define corresponding functions on the interval $I_r$. We use hat $\hat{\ }$ to emphasize that a certain quantity is defined on the reference interval.

The construction starts with the four functions

$$\begin{aligned}
\acute{\phi} &\colon \hat{I} \to \mathbb{R}, & \acute{\phi}(x) &= -x^3 + x^2, \\
\grave{\phi} &\colon \hat{I} \to \mathbb{R}, & \grave{\phi}(x) &= x^3 - 2x^2 + x, \\
\bar{\phi} &\colon \hat{I} \to \mathbb{R}, & \bar{\phi}(x) &= \acute{\phi}(x) + \grave{\phi}(x), \\
\tilde{\phi} &\colon \hat{I} \to \mathbb{R}, & \tilde{\phi}(x) &= \acute{\phi}(x) - \grave{\phi}(x),
\end{aligned} \tag{24}$$

defined on the reference interval $\hat{I}$ and shown in Figure 4. The corresponding functions on the interval $I_r$ are defined with the affine transform

$$\mathring{\phi}_r(x) := \begin{cases} \mathring{\phi} \circ T_r^{-1}(x), & x \in I_r \\ 0 & x \notin I_r, \end{cases} \tag{25}$$

extended by zero outside of $I_r$, for any $\mathring{\ } \in \{\acute{\ }, \grave{\ }, \bar{\ }, \tilde{\ }\}$.

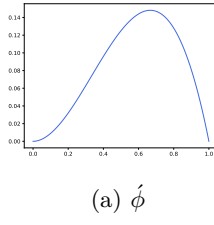

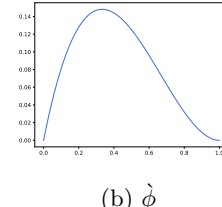

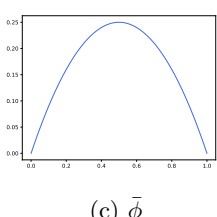

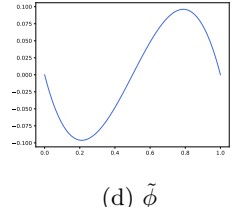

(a) $\acute{\phi}$        (b) $\grave{\phi}$        (c) $\bar{\phi}$        (d) $\tilde{\phi}$

Figure 4: Functions defined in (24).

From $\acute{\phi}_r$ and $\grave{\phi}_r$, we can construct all functions in $X^\perp$ that we need for the proof of the main results, as given in the next lemma.

**Lemma A.2.** *Let $\acute{\phi}_r$ and $\grave{\phi}_r$ be defined by* (25). *Then*

$$h_r \acute{\phi}_r'' - h_{r+1} \grave{\phi}_{r+1}'' \in X^\perp, \qquad\qquad r = 2, \ldots, \bar{m} - 1.$$

Although the second derivative might seem artificial at first, the function $\acute{\phi}_r$ will be more useful than $\acute{\phi}_r''$ down the road.

*Proof.* We abbreviate $\varphi_r := h_r \acute{\phi}_r'' - h_{r+1} \grave{\phi}_r''$. Clearly $\grave{\phi}_r''$ and $\acute{\phi}_r''$ are piecewise linear so that $\varphi_r \in X + \dot{X}$ and it is sufficient to show $\langle \varphi_r, H_s \rangle = 0$ for a basis $H_s$, $s = 0, \ldots, \bar{m}$ of $X$. We choose hat functions centered at $b_s$, i.e.

$$H_s(x) = \begin{cases} T_s^{-1}(x) & x \in I_s, \\ 1 - T_{s+1}^{-1}(x) & x \in I_{s+1}, \\ 0 & \text{else.} \end{cases}$$

Using the compact supports of $\acute{\phi}_r$, $\grave{\phi}_r$, $H_s$, the derivatives $T_r' = h_r$, $(T_r^{-1})' = h_r^{-1}$, the chain rule $\acute{\phi}_r'' = (\acute{\phi}'' \circ T_r^{-1})h_r^{-2}$ and transforming to the reference interval, we compute

$$\langle \varphi_r, H_{r-1} \rangle = h_r^{-1} \int_{I_r} \acute{\phi}'' \circ T_r^{-1}(x)(1 - T_r^{-1})(x)\,dx = \int_{\hat{I}} \acute{\phi}''(\hat{x})(1 - \hat{x})\,d\hat{x} = \int_0^1 (-6\hat{x} + 2)(1 - \hat{x})\,d\hat{x} = 0$$

and

$$\langle \varphi_r, H_{r+1} \rangle \;=\; h_{r+1}^{-1} \int_{I_{r+1}} \grave{\phi}'' \circ T_{r+1}^{-1}(x) T_{r+1}^{-1}(x)\, dx \;=\; \int_{\hat{I}} \grave{\phi}''(\hat{x}) \hat{x}\, d\hat{x} \;=\; \int_0^1 (6\hat{x} - 4)\hat{x}\, d\hat{x} \;=\; 0$$

and

$$\langle \varphi, H_r \rangle = h_r^{-1} \int_{I_r} \acute{\phi}'' \circ T_r^{-1}(x) T_r^{-1}(x)\, dx \;-\; h_{r+1}^{-1} \int_{I_{r+1}} \grave{\phi}'' \circ T_{r+1}^{-1}(x)(1 - T_{r+1}^{-1})(x)\, dx$$

$$= \int_0^1 (-6\hat{x} + 2)\hat{x}\, d\hat{x} - \int_0^1 (6\hat{x} - 4)(1 - \hat{x})\, d\hat{x} = (-1) - (-1) = 0$$

All other $\langle \varphi_r, H_s \rangle$ have non-overlapping support and therefore evaluate to zero. In conclusion $\langle \varphi_r, H_s \rangle = 0$ for a basis $H_s$ of $X^\perp$. Together with $\varphi_r \in X^\perp$, this concludes the proof.

$\square$

The following lemma is the cornerstone for showing equidistribution in Lemma 3.2 and Theorem 3.3.

**Lemma A.3.** *Let* $b_r$, $r \in [\bar{m}]$ *be cleaned critical breakpoints* (11), (12). *Let* $\acute{\phi}_r$, $\grave{\phi}_r$, $\bar{\phi}_r$, $\tilde{\phi}_r$ *be defined by* (25). *Then*

$$h_r \langle f'', \acute{\phi}_r \rangle_{I_r} = h_{r+1} \langle f'', \grave{\phi}_{r+1} \rangle_{I_{r+1}}.$$

Since $\acute{\phi}_r$ and $\grave{\phi}_r$ are non-negative bump functions, under the conditions of the main theorem, one can show that $h_r^{2 - \frac{1}{q}} \|f''\|_{L_q(I_r)} \sim h_r \langle f'', \acute{\phi}_r \rangle \sim h_r \langle f'', \grave{\phi}_r \rangle$ (with an argument analogous to Lemma A.27). Therefore, repeated application of the Lemma yields equidistribution

$$h_r^{2 - \frac{1}{q}} \|f''\|_{L_q(I_r)} \sim h_r \langle f'', \acute{\phi}_r \rangle = h_{r+1} \langle f'', \grave{\phi}_{r+1} \rangle \sim h_{r+1}^{2 - \frac{1}{q}} \|f''\|_{L_q(I_{r+1})}$$

between neighbouring intervals. However, repeated application for distant intervals $I_r$ and $I_s$ accumulates too much error. For the main equidistribution theorem we chain the identity of the lemma directly, which requires careful consideration of the difference $\tilde{\phi}_r = \acute{\phi}_r - \grave{\phi}_r$.

*Proof.* Recall from (23), that at critical points, we have $\langle f, v \rangle = 0$ for all $v \in X^\perp$. By Lemma A.2, the function $h_r \acute{\phi}''_r - h_{r+1} \grave{\phi}''_{r+1}$ is contained in $X^\perp$ and therefore we may substitute it for $v$ to obtain

$$\langle f, h_r \acute{\phi}''_r - h_{r+1} \grave{\phi}''_{r+1} \rangle = 0, \qquad \Leftrightarrow \qquad h_r \langle f, \acute{\phi}''_r \rangle = h_{r+1} \langle f, \grave{\phi}''_{r+1} \rangle. \qquad (26)$$

This shows the lemma, except that the two derivatives are on the wrong side of the inner product, which we correct with integration by parts:

$$h_r \langle f, \acute{\phi}''_r \rangle_{I_r} = -f(b_r) - h_r \langle f', \acute{\phi}'_r \rangle_{I_r} = -f(b_r) + h_r \langle f'', \acute{\phi}_r \rangle_{I_r}$$

where in the first step we have used that $\acute{\phi}'_r(b_{r-1}) = 0$ and $\acute{\phi}'_r(b_r) = \acute{\phi}' \circ T_r^{-1}(b_r) h_r^{-1} = -h_r^{-1}$ and in the second step that $\acute{\phi}_r$ is zero on the boundary, by the boundary values in Lemma A.4 in the technical supplements. Analogously, we obtain

$$h_{r+1} \langle f, \grave{\phi}''_{r+1} \rangle_{I_{r+1}} = -f(b_r) - h_{r+1} \langle f', \grave{\phi}'_{r+1} \rangle_{I_{r+1}} = -f(b_r) + h_{r+1} \langle f'', \grave{\phi}_{r+1} \rangle_{I_{r+1}}$$

Plugging into (26), we conclude that

$$-f(b_r) + h_r \langle f'', \acute{\phi}_r \rangle_{I_r} = -f(b_r) + h_{r+1} \langle f'', \grave{\phi}_{r+1} \rangle_{I_{r+1}},$$

which yields the lemma upon cancelling $f(b_r)$.

$\square$

### A.1.3 Technical Properties of $\acute{\phi}_r$, $\grave{\phi}_r$, $\bar{\phi}_r$, $\tilde{\phi}_r$

This section contains several technical properties of the functions $\acute{\phi}_r$, $\grave{\phi}_r$, $\bar{\phi}_r$, $\tilde{\phi}_r$ defined in (25).

**Lemma A.4.** *Let $\acute{\phi}_r$ and $\grave{\phi}_r$ be defined by (25). Then*

$$\acute{\phi}(0) = 0, \qquad \acute{\phi}(1) = 0, \qquad \acute{\phi}'(0) = 0, \qquad \acute{\phi}'(1) = -1,$$
$$\grave{\phi}(0) = 0, \qquad \grave{\phi}(1) = 0, \qquad \grave{\phi}'(0) = 1, \qquad \grave{\phi}'(1) = 0,$$

*Proof.* Follows directly from the explicit formulas for $\acute{\phi}$ and $\grave{\phi}$ in (24). $\qquad\square$

**Lemma A.5.** *Let $\acute{\phi}_r$ and $\grave{\phi}_r$ be defined by (25). Then for all $x \in \hat{I}$*

$$\acute{\phi}(x) \geq 0, \qquad\qquad\qquad\qquad \grave{\phi}(x) \geq 0.$$

*Proof.* $\acute{\phi}$ and $\grave{\phi}$ are third order polynomials for which the lemma follows by elementary computation.

$\square$

**Lemma A.6.** *Let $\acute{\phi}_r$ and $\grave{\phi}_r$ be defined by (25). Then*

$$\langle 1, \acute{\phi} \rangle_{I_r} = \frac{h_r}{12}, \qquad \langle T_r^{-1}, \acute{\phi} \rangle_{I_r} = \frac{h_r}{20}, \qquad \langle x, \acute{\phi} \rangle_{I_r} = \frac{h_r^2}{20} + \frac{h_r}{12} b_{r-1},$$
$$\langle 1, \grave{\phi} \rangle_{I_r} = \frac{h_r}{12}, \qquad \langle T_r^{-1}, \grave{\phi} \rangle_{I_r} = \frac{h_r}{30}, \qquad \langle x, \grave{\phi} \rangle_{I_r} = \frac{h_r^2}{30} + \frac{h_r}{12} b_{r-1}.$$

*Proof.* For any functions $\hat{v}$ and $\hat{\phi}$ defined on the reference interval $\hat{I}$ and corresponding functions $v := \hat{v} \circ T_r^{-1}$ and $\phi := \hat{\phi} \circ T_r^{-1}$ on the interval $I_r$, we have

$$\langle v, \phi \rangle_{I_r} = h_r \int_{\hat{I}} \hat{v}(\hat{x}) \hat{\phi}(\hat{x}) \, d\hat{x}.$$

Therefore, for $\hat{v} = 1$ and $\hat{\phi} = \acute{\phi}$ and $\hat{\phi} = \grave{\phi}$ we have

$$\langle 1, \acute{\phi}_r \rangle_{I_r} = h_r \langle 1, \acute{\phi} \rangle_{\hat{I}} = h_r \int_{\hat{I}} -\hat{x}^3 + \hat{x}^2 \, d\hat{x} = \frac{h_r}{12},$$
$$\langle 1, \grave{\phi}_r \rangle_{I_r} = h_r \langle 1, \grave{\phi} \rangle_{\hat{I}} = h_r \int_{\hat{I}} \hat{x}^3 - 2\hat{x}^2 + \hat{x} \, d\hat{x} = \frac{h_r}{12}.$$

Likewise, since the transformation of $\hat{x} \to \hat{x}$ to the interval $I_r$ is $T_r^{-1}$, we have

$$\langle T_r^{-1}, \acute{\phi}_r \rangle_{I_r} = h_r \langle \hat{x}, \acute{\phi} \rangle_{\hat{I}} = h_r \int_{\hat{I}} \hat{x}(-\hat{x}^3 + \hat{x}^2) \, d\hat{x} = \frac{h_r}{20},$$
$$\langle T_r^{-1}, \grave{\phi}_r \rangle_{I_r} = h_r \langle \hat{x}, \grave{\phi} \rangle_{\hat{I}} = h_r \int_{\hat{I}} \hat{x}(\hat{x}^3 - 2\hat{x}^2 + \hat{x}) \, d\hat{x} = \frac{h_r}{30}.$$

Finally, the function $x$ transformed to the reference interval is $T_r(\hat{x}) = (b_r - b_{r-1})\hat{x} + b_{r-1} = h_r\hat{x} + b_{r-1}$. Thus, together with the identities above

$$\langle x, \acute{\phi}_r \rangle_{I_r} = h_r \langle T_r, \acute{\phi} \rangle_{\hat{I}} = h_r^2 \langle \hat{x}, \acute{\phi} \rangle_{\hat{I}} + h_r b_{r-1} \langle 1, \acute{\phi} \rangle_{\hat{I}} = \frac{h_r^2}{20} + \frac{h_r}{12} b_{r-1},$$
$$\langle x, \grave{\phi}_r \rangle_{I_r} = h_r \langle T_r, \grave{\phi} \rangle_{\hat{I}} = h_r^2 \langle \hat{x}, \grave{\phi} \rangle_{\hat{I}} + h_r b_{r-1} \langle 1, \grave{\phi} \rangle_{\hat{I}} = \frac{h_r^2}{30} + \frac{h_r}{12} b_{r-1},$$

which completes the proof.

$\square$

### A.2 Equilibration for the Limit $h \to 0$

#### A.2.1 Results

We have seen in Section 2 that we can achieve optimal approximation rates by equilibrating the smoothness $\|f''\|_{L_{2/5}(I_r)}$ on all intervals $I_r$. In this section, we provide an argument that in the limit of small intervals $h_r \to 0$, the equidistribution and the critical point conditions yield the same adapted grids. This section is kept informal, put provides intuition and guidelines for a rigorous analysis for finite $h_r$ in Section A.3.

To provide meaningful limits, we consider the *grid size limit*

$$h(x) := \lim_{m \to \infty} mh_r,$$

where $h_r$ is the size of the interval $I_r$ that contains $x$ and the network width $m$ a normalization factor. On a uniform grid, this normalization yields $h(x) = |D|$. Throughout this section, we assume that the limit exists. We first consider the limit of equidistribution in the following lemma.

**Lemma A.7.** *Assume that the limit $h(x)$ exists. Let $f$ be smooth and let the intervals be equilibrated $\|f''\|_{L_{2/5}(I_r)} = \|f''\|_{L_{2/5}(I_s)}$ for all $r, s$ and for all $m$. Then for $m \to \infty$ the grid size limit satisfies*

$$h(x) = c|f''(x)|^{-2/5}$$

*for some constant c.*

This limit confirms the expectation that we want a fine grid wherever the function $f$ has little smoothness, here expressed by a large second derivative $|f''(x)|$. In comparison, the limiting grid of the critical points satisfy the following lemma.

**Lemma A.8** (Lemma 3.2 restated)**.** *Let $f$ be smooth and for every $m$ let $b_r$, $r \in [\bar{m}]$ be cleaned critical breakpoints (11), (12). If the limit $h(x)$ exists, it satisfies*

$$h(x) = c_I f''(x)^{-2/5},$$

*with possibly a different constant $c_I$ on each interval $I$ for which $f''(x)$ is non-zero.*

We observe that the grid size limit is identical to the density of the smoothness norm equidistribution in Lemma 3.2, up to a global factor on each interval for which $f''$ is non-zero. Thus, in the limit, critical points have a proper grid distribution on every strictly convex or concave stretch of the target function $f$, but may be imbalanced between these stretches.

The proof relies on the observation that in the limit the grid size satisfies the ODE in the following lemma.

**Lemma A.9.** *Let $f$ be smooth and for every $m$ let $b_r$, $r \in [\bar{m}]$ be cleaned critical breakpoints (11), (12). If the limit $h(x)$ exists, it satisfies the differential equation*

$$[h(x)^2 f''(x)]' = \frac{1}{5} h(x)^2 f'''(x).$$

The proofs of all lemmas are given in the following section.

#### A.2.2 Proofs

We first prove the limit for norm equidistribution.

*Proof of Lemma 3.2.* Since $f$ is smooth, we have

$$m^{5/2} \|f''\|_{L_{2/5}(I_r)} = (mh_r)^{5/2} \left( \frac{1}{h_r} \int_{I_r} |f''(x)|^{2/5} \, dx \right)^{5/2} \to h(x)^{5/2} |f''(x)|.$$

Since by equidistribution the left hand side is independent of the interval $I_r$ containing $x$, the right hand side is independent of $x$ and thus

$$h(x)^{5/2}|f''(x)| = \text{const} \qquad\qquad \Rightarrow \qquad\qquad h(x) = \text{const}|f''(x)|^{-2/5},$$

which concludes the proof.

$\square$

To prove the analogous result for critical points, recall that from Lemma A.3 that $h_r\langle f'', \acute{\phi}_r\rangle_{I_r} = h_{r+1}\langle f'', \grave{\phi}_{r+1}\rangle_{I_{r+1}}$. We show that in the limit this reduces to an ODE for the grid size $h(x)$. To this end, we first need some technical lemmas.

**Lemma A.10.** *Let $v$ be smooth and $\acute{\phi}_r$, $\grave{\phi}_r$ be defined by (25). Then*

$$\lim_{m\to\infty} \frac{1}{h_r}\langle v, \acute{\phi}_r\rangle_{I_r} = \frac{1}{12}v(x),$$
$$\lim_{m\to\infty} \frac{1}{h_r^2}\langle v, \acute{\phi}_r - \grave{\phi}_r\rangle_{I_r} = \frac{1}{60}v'(x).$$

*Proof.* Since $v$ is smooth and $\acute{\phi}_r$ non-negative (Lemma A.5), by the mean value theorem and normalization $\langle 1, \acute{\phi}\rangle_{I_r} = h_r/12$ (Lemma A.6), we have

$$\frac{1}{h_r}\langle v, \acute{\phi}_r\rangle_{I_r} = v(\xi)\frac{1}{h_r}\langle 1, \acute{\phi}_r\rangle_{I_r} = \frac{1}{12}v(\xi)$$

for some $\xi \in I_r$. In the limit $m \to \infty$ and intervals $I_r$ that contain $x$, this yields the first formula of the lemma.

To show the second limit, define

$$\Phi(x) := \int_0^x \acute{\phi}(y) - \grave{\phi}(y)\, dy = -\frac{1}{2}(x^4 - 2x^3 + x^2)$$

and $\Phi_r := \Phi \circ (T_r)^{-1}$. Then $\Phi_r' = \Phi' \circ T_r^{-1}h_r^{-1} = (\acute{\phi}_r - \grave{\phi}_r)h_r^{-1}$. Furthermore, $\Phi$ has boundary values $\Phi(0) = \Phi(1) = 0$, is non-positive on $\hat{I}$ and

$$\langle 1, \Phi_r\rangle_{I_r} = h_r \int_{\hat{I}} \Phi(\hat{x})\, d\hat{x} = -\frac{h_r}{60}$$

by transforming to the reference interval with $T_r$ and $T_r' = h_r$. Using integration by parts, $\Phi \le 0$ and the mean value theorem, it follows that

$$\frac{1}{h_r^2}\langle v, \acute{\phi}_r - \grave{\phi}_r\rangle_{I_r} = -\frac{1}{h_r}\langle v', \Phi_r\rangle_{I_r} = -v'(\xi)\frac{1}{h_r}\langle 1, \Phi_r\rangle_{I_r} = v'(\xi)\frac{1}{60}.$$

Taking the limit $m \to \infty$, this yields the second formula of the lemma.

$\square$

Next, we show that the grid size limit satisfies the ODE in Lemma A.9.

*Proof of Lemma A.9.* By a telescopic sum, we have

$$h_s\langle f'', \acute{\phi}_s\rangle_{I_s} - h_r\langle f'', \acute{\phi}_r\rangle_{I_r} = \sum_{k=r+1}^{s} h_k\langle f'', \acute{\phi}_k\rangle_{I_k} - h_{k-1}\langle f'', \acute{\phi}_{k-1}\rangle_{I_{k-1}}.$$

Since the intervals $I_r$ originate from are a critical point, by Lemma A.3 we have $h_{k-1}\langle f'', \acute{\phi}_{k-1}\rangle_{I_{k-1}} = h_k\langle f'', \grave{\phi}_k\rangle_{I_k}$ and therefore

$$h_s\langle f'', \acute{\phi}_s\rangle_{I_s} - h_r\langle f'', \acute{\phi}_r\rangle_{I_r} = \sum_{k=r+1}^{s} h_k\langle f'', \acute{\phi}_k\rangle_{I_k} - h_k\langle f'', \grave{\phi}_k\rangle_{I_k}$$

$$= \sum_{k=r+1}^{s} h_k\langle f'', \acute{\phi}_k - \grave{\phi}_k\rangle_{I_k}.$$

Multiplying by $m^2$ yields

$$(mh_s)^2 \frac{1}{h_s}\langle f'', \acute{\phi}_s\rangle_{I_s} - (mh_r)^2 \frac{1}{h_r}\langle f'', \acute{\phi}_r\rangle_{I_r} = \sum_{k=r+1}^{s} (mh_k)^2 \frac{1}{h_k^2}\langle f'', \acute{\phi}_k - \grave{\phi}_k\rangle_{I_k} h_k.$$

By Lemma A.10, in the limit $m \to \infty$ this converges to

$$\frac{1}{12} h(x)^2 f''(x) - \frac{1}{12} h(y)^2 f''(y) = \frac{1}{60} \int_y^x h(z)^2 f'''(z)\, dz,$$

where we have used that the terms in the Riemann sum are constant on each interval $I_r$ so that the extra $h_r$ at the end converges to $dz$. Multiplying by 12 and differentiation with respect to $x$ yields the lemma.

$\square$

It remains to solve the ODE in the last Lemma.

*Proof of Lemma 3.2, A.8.* We abbreviate $g := f''$. By Lemma A.9, the grid size limit $h$ satisfies the ODE

$$[h^2 g]' = \frac{1}{5} h^2 g',$$

which is first order linear in $h^2$, whenever $g(x) \neq 0$. To solve it, define the integrating factor $\mu$ by the ODE

$$g\mu' = -\frac{1}{5} g'\mu,$$

with the explicit solution

$$g\mu' = -\frac{1}{5} g'\mu \qquad \Rightarrow \qquad \frac{\mu'}{\mu} = -\frac{1}{5}\frac{g'}{g} \qquad \Rightarrow \qquad \ln(\mu)' = -\frac{1}{5}\ln(g)' \qquad \Rightarrow \qquad \mu = g^{-1/5},$$

up to a global multiplicative factor. Multiplying the original ODE with $\mu$, we obtain

$$[h^2 g]'\mu = \frac{1}{5} h^2 g'\mu = -h^2 g\mu' \qquad \Rightarrow \qquad [h^2 g]'\mu + h^2 g\mu' = 0 \qquad \Rightarrow \qquad [h^2 g\mu]' = 0.$$

Hence $h^2 g\mu$ is constant. Plugging in $g = f''$ and the explicit solution $\mu = g^{-1/5}$ yields $h^2(f'')^{4/5} = c$ for some constant $c$. Solving for $h$ yields the lemma.

$\square$

## A.3 Equilibrium for Finite $h$

In this section, we prove the main equidistribution theorem, restated here for convenience:

**Theorem A.11** (Theorem 3.3, restated)**.** *Let $\theta$ be a critical point* (11)*, with cleaned breakpoints in ascending order 12. For $r, s \in \{2, \ldots, \bar{m}\}$, let $\mathcal{I} = \{r, r+1, \ldots, s\}$ be a set of consecutive neurons with $D_\mathcal{I} := \bigcup_{k \in \mathcal{I}} I_k$ and*

$$\max\left\{ h_{k+}^{\frac{1}{2}-\frac{1}{p}}\|f^{(3)}\|_{L_p(I_{k+})}, \; h_{k+}^{1-\frac{1}{q}}\|f^{(4)}\|_{L_q(I_{k+})}, \right\} \leq C \min_{x \in I_{k+}} |f''(x)| \tag{27}$$

*for some $1 < q, p \leq \infty$ and some sufficiently small constant $C > 0$ independent of $f$ and $h_k$. Then for $l, k \in D_\mathcal{I}$ we have equidistribution*

$$\|f''\|_{L_{2/5}(I_l)} \sim \|f''\|_{L_{2/5}(I_k)}.$$

### A.3.1 Notations

Throughout this section, we abbreviate

$$\acute{a}_r(v) := h_r^{-1}\langle v, \acute{\phi}_r\rangle_{I_r}, \qquad\qquad \grave{a}_r(v) := h_r^{-1}\langle v, \grave{\phi}_r\rangle_{I_r}, \qquad (28)$$

$$\bar{a}_r(v) := h_r^{-1}\langle v, \bar{\phi}_r\rangle_{I_r}, \qquad\qquad \tilde{a}_r(v) := h_r^{-1}\langle v, \tilde{\phi}_r\rangle_{I_r} \qquad (29)$$

and in case $v = f''$, even shorter

$$\acute{a}_r := \acute{a}_r(f''), \qquad \grave{a}_r := \grave{a}_r(f''), \qquad \bar{a}_r := \bar{a}_r(f''), \qquad \tilde{a}_r := \tilde{a}_r(f''). \qquad (30)$$

We will also repeatedly use the integrating factor

$$\mu_r := |\bar{a}_r|^\alpha, \qquad\qquad \alpha := -\frac{1}{5}. \qquad (31)$$

analogous to the one that was used in the solution of the ODE from Lemma A.9, in the infinite width limit.

### A.3.2 Overview

Recall from Lemma A.9 that for $m \to \infty$, the grid size limit $h(x)$ and the integrating factor $\mu(x)$ satisfy the two ODEs

$$[h^2 f'']^2 = \frac{1}{5}h^2 f''', \qquad\qquad f''\mu' = -\frac{1}{5}f'''\mu,$$

respectively. It follows that

$$\begin{aligned} [h^2 f''\mu]' &= [h^2 f'']'\mu + h^2 f''\mu' \\ &= \frac{1}{5}h^2 f'''\mu - \frac{1}{5}h^2 f'''\mu \\ &= 0, \end{aligned} \qquad (32)$$

where in the first step we have used the product rule and in the third the two ODEs for $h$ and $\mu$. For finite $h_r$, we follow a similar argument:

1. Lemma A.12 replaces the derivative on the left hand side with a difference and the zero on the right hand side with perturbation terms $(I) - (IV)$ that we prove to be small in subsequent sections.

   As result, the terms $h_r^2 \bar{a}_r \mu_r$ are almost constant between neighbouring intervals.

2. Lemmas A.13 and A.14 bound error accumulation when comparing $h_r^2 \bar{a}_r \mu_r = [h_r^{\frac{5}{2}}\bar{a}_s]^{\frac{4}{5}}$ over multiple intervals, resulting in equidistribution of this quantity.

3. Theorem 3.3 then follows from $[h_r^{\frac{5}{2}}\bar{a}_s]^{\frac{4}{5}} \sim \|f''\|_{L_{2/5}(I_r)}^{\frac{4}{5}}$, by Lemma A.27 at the end of this section.

### A.3.3 Proof of the Main Result

The assumptions for the main theorems confine the results to regions where $f$ is convex or concave. For the time being, we make this assumption explicit by assuming $\bar{a}_r \geq 0$, which will be removed later.

**Lemma A.12.** *Let $b_r$, $r \in [\bar{m}]$ be cleaned critical breakpoints (11), (12). Let $\bar{a}_r$, $\tilde{a}_r$ and $\mu_r$ be defined by (28), (31), $\bar{a}_r \geq 0$ and $\alpha := -1/5$. Then*

$$h_{r+1}^2 \bar{a}_{r+1}\mu_{r+1} - h_r^2 \bar{a}_r \mu_r = (I) + (II) + (III) + (IV),$$

*with*

$$\begin{aligned} (I) &= h_r^2[\tilde{a}_{r+1} + \tilde{a}_k + \alpha[\bar{a}_{r+1} - \bar{a}_r]]\mu_{r+1}, \\ (II) &= [h_{r+1}^2 - h_r^2]\tilde{a}_{r+1}\mu_{r+1}, \\ (III) &= \alpha^2 h_r^2 \bar{a}_r^{\alpha-1}[\bar{a}_{r+1} - \bar{a}_r]^2, \\ (IV) &= \alpha h_r^2[\bar{a}_{r+1} - \bar{a}_r]R_r + h_r^2 \bar{a}_r R_r. \end{aligned}$$

*and for some $0 \leq \xi_r \leq 1$*

$$R_r := \alpha(\alpha - 1)[\xi_r \bar{a}_r + (1 - \xi_r)\bar{a}_{r+1}]^{\alpha-2}[\bar{a}_{r+1} - \bar{a}_r]^2.$$

*Proof.* We mimic the steps in the motivation (32), starting with the product rule:

$$h_{r+1}^2 \bar{a}_{r+1} \mu_{r+1} - h_r^2 \bar{a}_r \mu_r = [h_{r+1}^2 \bar{a}_{r+1} - h_r^2 \bar{a}_r]\mu_{r+1} + h_r^2 \bar{a}_r [\mu_{r+1} - \mu_r]. \tag{33}$$

The next step in the motivation is to invoke the ODEs for $h$ and $\mu$. The former is based on Lemma A.3, which we can invoke directly (twice in the second step below) together with the observation that $h_r^2 \acute{a}_r = h_r \langle f'', \acute{\phi}_r \rangle_{I_r}$, etc., to obtain

$$\begin{aligned}
h_{r+1}^2 \bar{a}_{r+1} - h_r^2 \bar{a}_k &= (h_{r+1}^2 \acute{a}_{r+1} - h_k^2 \acute{a}_k) + (h_{r+1}^2 \grave{a}_{r+1} - h_k^2 \grave{a}_k) \\
&= (h_{r+1}^2 \acute{a}_{r+1} - h_{r+1}^2 \grave{a}_{r+1}) + (h_r^2 \acute{a}_r - h_k^2 \grave{a}_k) \\
&= h_{r+1}^2 \tilde{a}_{r+1} + h_r^2 \tilde{a}_k.
\end{aligned}$$

For the second term, we don't invoke the $\mu$ ODE directly, but instead compute the derivative, or rather difference, using the explicit formula $\mu_r = \bar{a}_r^\alpha$. Applying a Taylor expansion for $z \to z^\alpha$, we obtain

$$\mu_{r+1} - \mu_r = \bar{a}_{r+1}^\alpha - \bar{a}_r^\alpha = \alpha \bar{a}_r^{\alpha-1}[\bar{a}_{r+1} - \bar{a}_r] + R_r \tag{34}$$

and Taylor remainder

$$R_r := \alpha(\alpha - 1)[\xi_r \bar{a}_r + (1 - \xi_r)\bar{a}_{r+1}]^{\alpha-2}[\bar{a}_{r+1} - \bar{a}_r]^2 \tag{35}$$

for some $0 < \xi_r < 1$. Plugging these identities into (33) and using $\bar{a}_r^\alpha = \mu_r$, we obtain

$$\begin{aligned}
h_{r+1}^2 \bar{a}_{r+1} \mu_{r+1} - h_r^2 \bar{a}_r \mu_r &= [h_{r+1}^2 \tilde{a}_{r+1} + h_r^2 \tilde{a}_k]\mu_{r+1} + h_r^2 \bar{a}_r [\alpha \bar{a}_r^{\alpha-1}[\bar{a}_{r+1} - \bar{a}_r] + R_r] \\
&= [h_{r+1}^2 \tilde{a}_{r+1} + h_r^2 \tilde{a}_k]\mu_{r+1} + \alpha h_r^2 \mu_r [\bar{a}_{r+1} - \bar{a}_r] + h_r^2 \bar{a}_r R_r.
\end{aligned}$$

In the continuous motivation (32), the terms on the right hand side cancel to zero. In the discrete case, we rearrange the right hand side into summands that we prove to be small later:

$$\begin{aligned}
h_{r+1}^2 \bar{a}_{r+1} \mu_{r+1} - h_r^2 \bar{a}_r \mu_r &= [h_{r+1}^2 - h_r^2]\tilde{a}_{r+1}\mu_{r+1} \\
&\quad + h_r^2 [\tilde{a}_{r+1} + \tilde{a}_k]\mu_{r+1} + \alpha h_r^2 \mu_{r+1}[\bar{a}_{r+1} - \bar{a}_r] \\
&\quad - \alpha h_r^2 [\mu_{r+1} - \mu_r][\bar{a}_{r+1} - \bar{a}_r] + h_r^2 \bar{a}_r R_r \\
&= [h_{r+1}^2 - h_r^2]\tilde{a}_{r+1}\mu_{r+1} \\
&\quad + h_r^2 [\tilde{a}_{r+1} + \tilde{a}_k + \alpha[\bar{a}_{r+1} - \bar{a}_r]]\mu_{r+1} \\
&\quad - \alpha h_r^2 [\mu_{r+1} - \mu_r][\bar{a}_{r+1} - \bar{a}_r] + h_r^2 \bar{a}_r R_r.
\end{aligned}$$

We will see later that the third but last and last lines contain small perturbation terms and the second but last line is zero for $f'' \in \mathbb{P}^1$ (Lemma A.16) and close to zero for general $f''$. For now, we eliminate the difference $\mu_{r+1} - \mu_r$ by Taylor expansion (34), (35) to obtain

$$\alpha h_r^2 [\mu_{r+1} - \mu_r][\bar{a}_{r+1} - \bar{a}_r] = \alpha^2 h_r^2 \bar{a}_r^{\alpha-1}[\bar{a}_{r+1} - \bar{a}_r]^2 + \alpha h_r^2 [\bar{a}_{r+1} - \bar{a}_r]R_r$$

and therefore

$$\begin{aligned}
h_{r+1}^2 \bar{a}_{r+1} \mu_{r+1} - h_r^2 \bar{a}_r \mu_r &= [h_{r+1}^2 - h_r^2]\tilde{a}_{r+1}\mu_{r+1} \\
&\quad + h_r^2 [\tilde{a}_{r+1} + \tilde{a}_k + \alpha[\bar{a}_{r+1} - \bar{a}_r]]\mu_{r+1} \\
&\quad - \alpha^2 h_r^2 \bar{a}_r^{\alpha-1}[\bar{a}_{r+1} - \bar{a}_r]^2 + \alpha h_r^2 [\bar{a}_{r+1} - \bar{a}_r]R_r + h_r^2 \bar{a}_r R_r
\end{aligned}$$

This concludes the proof, upon reordering terms.

$\square$

The previous lemma shows that $h_r^2 \bar{a}_r \mu_r$ is comparable on two neighbouring intervals $I_r$ and $I_{r+1}$. Applying the argument repeatedly, allows us to compare $h_r^2 \bar{a}_r \mu_r$ across multiple intervals. The following lemma bounds the compound error.

**Lemma A.13.** *Let $z_k \in \mathbb{R}$ and $h_k \geq 0$ for $k = 1, \ldots, m$. Assume*

$$|z_{k+1} - z_k| \leq c_d[h_k z_k + h_{k+1} z_{k+1}]$$

*and the conditions*

$$c_d h_k \leq \frac{1}{2}, \qquad 1 + 2c_d C_d \sum_{k=1}^m h_k \leq C_d, \qquad \frac{1}{C_d} \leq 1 - 2c_d \sum_{k=1}^m h_k$$

*for the two constants $c_d, C_d \geq 0$. Then for all $r, s \in [m]$ we have*

$$\frac{1}{C_d} z_r \leq z_s \leq C_d z_r.$$

*Proof.* We first show that $z_k$ does not change sign. To this end, assume $z_{k+1} \geq 0$ and $z_k \leq 0$. Then, by assumption

$$|z_{k+1}| + |z_k| \leq c_d[h_k|z_k| + h_{k+1}|z_{k+1}|] \leq \frac{1}{2}|z_{k+1}| + |z_k|,$$

which directly implies $z_k = z_{k+1} = 0$. Therefore, in the following we assume without loss of generality that $z_k \geq 0$.

By symmetry it suffices to show $z_s \leq C_d z_r$. To this end, assume without loss of generality $r \leq s$ and by induction that the statement is true for all $r \leq k \leq s - 1$. We show the statement for $k = s$. With a telescopic sum, we have

$$z_s - z_r = \sum_{k=r}^{s-1} z_{k+1} - z_k \leq c_d \sum_{k=r}^{s-1} h_k z_k + h_{k+1} z_{k+1} \leq 2c_d \sum_{k=r}^s h_k z_k$$

$$\leq 2c_d \left( \sum_{k=r}^s h_k \right) \max\{z_s, \max_{r \leq k < s} z_k\} \leq 2c_d \left( \sum_{k=r}^s h_k \right) \max\{z_s, C_d z_r\},$$

where in the second step we have used the first assumption of the lemma, in the third an index shift on the $h_{k+1} z_{k+1}$ summands and in the last the induction hypothesis.

We proceed with the two options for the maximum separately. In case $\max\{z_s, C_d z_r\} = z_s$, we have

$$z_s - z_r \leq 2c_d \left( \sum_{k=r}^s h_k \right) z_s \qquad \Rightarrow \qquad z_s \leq \left[ 1 - 2c_d \left( \sum_{k=r}^s h_k \right) \right]^{-1} z_r \leq C_d z_r,$$

where we have solved the first inequality for $z_s$ and then estimated the bracket by the given assumptions.

By the same reasoning, in the case $\max\{z_s, C_d z_r\} = C_d z_r$ we have

$$z_s - z_r \leq 2c_d \left( \sum_{k=r}^s h_k \right) C_d z_r \qquad \Rightarrow \qquad z_s \leq \left[ 1 + 2c_d C_d \left( \sum_{k=r}^s h_k \right) \right] z_r \leq C_d z_r,$$

Thus, in any case we have $z_s \leq C_d z_r$ and the lemma follows by induction.

$\square$

Combining the last two lemmas shows that $h_r^2 \bar{a}_r \mu_r$ is equidistributed across multiple intervals $I_r$. This requires us to bound the terms $(I) - (IV)$, which is technical and deferred to Section A.3.5 later.

**Lemma A.14.** *Let $b_r$, $r \in [\bar{m}]$ be cleaned critical breakpoints (11), (12). Assume that*

$$\max\left\{ h_{k+}^{\frac{1}{2}-\frac{1}{p}}\|f^{(3)}\|_{L_p(I_{k+})}, \; h_{k+}^{1-\frac{1}{q}}\|f^{(4)}\|_{L_q(I_{k+})}, \right\} \leq C \min_{x \in I_{k+}} |f''(x)|$$

*for some $1 < q, p \leq \infty$ and some sufficiently small constant $C > 0$, independent of $f$ and $h_r$, and $r$ contained in some consecutive indices $\mathcal{I} \subset [\bar{m}]$. Then*

$$h_s^2|\bar{a}_s|\mu_s \sim h_r^2|\bar{a}_r|\mu_r$$

*for all $r, s \in \mathcal{I}$.*

*Proof.* First note that by Lemma A.20 in the technical supplements we have $\bar{a}_r \sim \bar{a}_{r+1}$ so that they cannot change sign. Upon eventually replacing $f$ with $-f$, we may assume without loss of generality that $\bar{a}_r \geq 0$. Then, the result follows from Lemma A.13 with the choice $z_k = h_k^2 \bar{a}_k \mu_k$. To prove its assumptions, we have to show

$$h_{r+1}^2\bar{a}_{r+1}\mu_{r+1} - h_r^2\bar{a}_r\mu_r \leq c_d(h_r^2\bar{a}_r\mu_r)h_r + c_d(h_{r+1}^2\bar{a}_{r+1}\mu_{r+1})h_{r+1}$$

for some sufficiently small $c_d$ so that the lemmas restrictions on the constants are satisfied. By Lemma A.12, we have

$$h_{r+1}^2\bar{a}_{r+1}\mu_{r+1} - h_r^2\bar{a}_r\mu_r = (I) + (II) + (III) + (IV),$$
$$\lesssim C(h_r^2\bar{a}_r\mu_r)h_r + C(h_{r+1}^2\bar{a}_{r+1}\mu_{r+1})h_{r+1}$$

for some terms $(I) - (IV)$ that are bounded by Lemmas A.23, A.24, A.25, A.26. These lemmas require

$$h_{r+}^{\frac{1}{2}-\frac{1}{p}}\|f^{(3)}\|_{L_p(I_{r+})} \leq C \min_{x \in I_{r+}} |f''(x)|,$$
$$h_{r+}^{1-\frac{1}{q}}\|f^{(4)}\|_{L_q(I_{r+})} \leq C \min_{x \in I_{r+}} |f''(x)|,$$

possibly with different $1 \leq p, q \leq \infty$ and for sufficiently small constant $C$. This matches the given assumption and concludes the proof.

$\square$

The technical Lemma A.27 below shows that $h_r^{\frac{5}{2}}|\bar{a}_s| \sim \|f''\|_{L_{2/5}(I_r)}$, which allows us to conclude the proof of the main Theorem 3.3.

*Proof of Theorem 3.3.* From Lemma A.14 together with the definition $\mu_r := |\bar{a}_r|^{-1/5}$, we have the equidistribution

$$h_s^2|\bar{a}_s|^{\frac{4}{5}} \sim h_r^2|\bar{a}_r|^{\frac{4}{5}}$$

and by Lemma A.27, we have the equivalence

$$h_r^2|\bar{a}_s|^{\frac{4}{5}} = [h_r^{\frac{5}{2}}|\bar{a}_s|]^{\frac{4}{5}} \sim \|f''\|_{L_{2/5}(I_r)}^{\frac{4}{5}}.$$

Combining these equivalences, the norms $\|f''\|_{L_{2/5}(I_r)}^{\frac{4}{5}}$ are equilibrated and the result follows.

$\square$

### A.3.4 Technical Lemmas

This Section contains a collection of technical lemmas that are used to bound $(I) - (IV)$ in Lemma A.12.

**Lemma A.15.** *Let $\acute{\phi}_r$, $\grave{\phi}_r$, $\bar{\phi}_r$, $\tilde{\phi}_r$ be defined by (25) and $\bar{a}_r$ by (28). Then*

$$\tilde{a}_r(1) = 0, \qquad\qquad\qquad \tilde{a}_r(x) = \frac{h_r}{60},$$

$$\bar{a}_r(1) = \frac{1}{6}, \qquad\qquad\qquad \bar{a}_r(x) = \frac{h_r}{12} + \frac{1}{6}b_{r-1},$$

$$\bar{a}_{r+1}(1) - \bar{a}_r(1) = 0, \qquad\qquad\qquad \bar{a}_{r+1}(x) - \bar{a}_r(x) = \frac{h_{r+1} + h_r}{12}.$$

*Proof.* From $\tilde{\phi}_r = \acute{\phi}_r - \grave{\phi}_r$ and Lemma A.6, we have

$$\langle 1, \tilde{\phi}_r \rangle_{I_r} = \langle 1, \acute{\phi}_r \rangle_{I_r} - \langle 1, \grave{\phi}_r \rangle_{I_r} = \frac{h_r}{12} - \frac{h_r}{12} = 0,$$

$$\langle x, \tilde{\phi}_r \rangle_{I_r} = \langle x, \acute{\phi}_r \rangle_{I_r} - \langle x, \grave{\phi}_r \rangle_{I_r} = \left( \frac{h_r^2}{20} + \frac{h_r}{12}b_{r-1} \right) - \left( \frac{h_r^2}{30} + \frac{h_r}{12}b_{r-1} \right) = \frac{h_r^2}{60}.$$

Dividing by $h_r$ and plugging in the definition of $\tilde{a}_r$ on the left hand side shows the first two identities of the lemma. Likewise, with the definition $\bar{\phi}_r = \acute{\phi}_r + \grave{\phi}_r$, we have

$$\langle 1, \bar{\phi}_r \rangle_{I_r} = \langle 1, \acute{\phi}_r \rangle_{I_r} + \langle 1, \grave{\phi}_r \rangle_{I_r} = \frac{h_r}{12} + \frac{h_r}{12} = \frac{h_r}{6},$$

$$\langle x, \bar{\phi}_r \rangle_{I_r} = \langle x, \acute{\phi}_r \rangle_{I_r} + \langle x, \grave{\phi}_r \rangle_{I_r} = \left( \frac{h_r^2}{20} + \frac{h_r}{12}b_{r-1} \right) + \left( \frac{h_r^2}{30} + \frac{h_r}{12}b_{r-1} \right) = \frac{h_r^2}{12} + \frac{h_r}{6}b_{r-1},$$

which shows the second two identities of the lemma. It follows that

$$h_{r+1}^{-1}\langle 1, \bar{\phi}_{r+1} \rangle_{I_{r+1}} - h_r^{-1}\langle 1, \bar{\phi}_r \rangle_{I_r} = \frac{1}{6} - \frac{1}{6} = 0,$$

$$h_{r+1}^{-1}\langle x, \bar{\phi}_{r+1} \rangle_{I_{r+1}} - h_r^{-1}\langle x, \bar{\phi}_r \rangle_{I_r} = \left( \frac{h_{r+1}}{12} + \frac{1}{6}b_r \right) - \left( \frac{h_r}{12} + \frac{1}{6}b_{r-1} \right)$$

$$= \frac{h_{r+1} - h_r}{12} + \frac{h_r}{6} = \frac{h_{r+1} + h_r}{12},$$

where we have used that $b_r - b_{r-1} = h_r$. Again, plugging in the definitions of $\bar{a}_r$ on the left hand side shows the remaining identities of the lemma.

$\square$

**Lemma A.16.** *Let $\bar{a}_r$ and $\tilde{a}_r$ be defined by (28) and $\alpha := -1/5$. Then*

$$\tilde{a}_{r+1}(p) + \tilde{a}_r(p) + \alpha[\bar{a}_{r+1}(p) - \bar{a}_r(p)] = 0.$$

*for all linear $p \in \mathbb{P}^1$.*

*Proof.* Since $\bar{a}_r(\cdot)$ and $\tilde{a}_r(\cdot)$ are linear, it suffices to show the lemma for $p = 1$ and $p = x$. For the former by Lemma A.15 we have

$$\tilde{a}_{r+1}(1) + \tilde{a}_r(1) + \alpha[\bar{a}_{r+1}(1) - \bar{a}_r(1)] = 0 + 0 - \alpha[0] = 0.$$

For the latter, we have

$$\tilde{a}_{r+1}(x) + \tilde{a}_r(x) + \alpha[\bar{a}_{r+1}(x) - \bar{a}_r(x)] = \frac{h_{r+1}}{60} + \frac{h_r}{60} + \alpha\left[ \frac{h_{r+1} + h_r}{12} \right] = 0,$$

because $\alpha = -1/5$.

$\square$

**Lemma A.17.** *Let $\acute{\phi}_r$ and $\grave{\phi}_r$ be defined by (25). Then for any $0 < p < \infty$ and integer $s \geq 0$ the $L_p$ norms of the $s$-th derivatives are bounded by*

$$\|\acute{\phi}_r^{(s)}\|_{L_p(I_r)} \sim h_r^{\frac{1}{p}-s}, \qquad\qquad \|\grave{\phi}_r^{(s)}\|_{L_p(I_r)} \sim h_r^{\frac{1}{p}-s},$$

*with constants that depend on $p$ and $s$.*

*Proof.* On the reference interval $\hat{I}$, we have $\|\acute{\phi}^{(s)}\|_{L_p(\hat{I})} \sim 1$ form some constants that depend on $p$ and $s$. Hence, we only need to check the scaling for the transform to the interval $I_r$ by integral substitution:

$$\|\acute{\phi}_r^{(s)}\|_{L_p(I_r)}^p = \int_{I_r} |\acute{\phi}^{(s)} \circ T_r^{-1}(x) h_r^{-s}|^p \, dx = h_r^{1-sp} \int_{I_r} |\acute{\phi}(\hat{x})|^p \, d\hat{x} \sim h_r^{1-sp}.$$

Taking the $p$-th root shows the claimed equivalences for $\acute{\phi}_r$. The result for $\grave{\phi}_r$ follows analogously.

$\square$

**Lemma A.18.** *Let $\bar{a}_r(v)$ and $\tilde{a}_r(v)$ be defined by (28) and $1 \leq p \leq \infty$. Define the joint interval $I_{r+} := I_r \cup I_{r+1}$ of size $h_{r+} := |I_{r+}|$. Then*

1. $|\bar{a}_r(v)| \leq h_r^{-\frac{1}{p}}\|v\|_{L_p(I_r)}.$

2. $|\tilde{a}_r(v)| \leq h_r^{1-\frac{1}{p}}\|v'\|_{L_p(I_r)}.$

3. $|\bar{a}_{r+1}(v) - \bar{a}_r(v)| \leq h_{r+}\left(h_{r+1}^{-\frac{1}{p}} + h_r^{-\frac{1}{p}}\right)\|v'\|_{L_p(I_{r+})}.$

4. $\bar{a}_r(v) \geq \frac{1}{6}\min_{x \in I_r} v(x)$ and $\square_r(v) \geq \frac{1}{12}\min_{x \in I_r} v(x)$ for $\square \in \{\acute{a}, \grave{a}\}.$

*Proof.* Throughout the proof define $q$ by $1/p + 1/q = 1$ so that $-1 + 1/q = -1/p$.

1. By Hölder's inequality and Lemma A.17 we have

$$|\bar{a}_r(v)| = h_r^{-1}\langle v, \acute{\phi}_r + \grave{\phi}_r\rangle_{I_r} \leq h_r^{-1}\|v\|_{L_p(I_r)}\|\acute{\phi}_r + \grave{\phi}_r\|_{L_q(I_r)}$$
$$\leq h_r^{-1}\|v\|_{L_p(I_r)}h_r^{\frac{1}{q}} \leq h_r^{-\frac{1}{p}}\|v\|_{L_p(I_r)}.$$

2. By Lemma A.15 we have $\tilde{a}_r(1) = 0$. Hence, if $c$ is the best $L_p$ constant approximation to $v$, with standard direct approximation inequalities ((41) in the supplementary material), we obtain

$$|\tilde{a}_r(v)| = |\tilde{a}_r(v - c)| = h_r^{-1}\langle v - c, \acute{\phi}_r - \grave{\phi}_r\rangle_{I_r}$$
$$\leq h_r^{-1}\|v - c\|_{L_p(I_r)}\|\acute{\phi}_r - \grave{\phi}_r\|_{L_q(I_r)} \leq h_r^{-1}h_r\|v'\|_{L_p(I_r)}h_r^{\frac{1}{q}}$$
$$\leq h_r^{1-\frac{1}{p}}\|v'\|_{L_p(I_r)}.$$

3. By Lemma A.15 we have $\bar{a}_{r+1}(1) - \bar{a}_r(1) = 0$. Hence, if $c$ is the best $L_p$ constant approximation to $v$ on the joint interval $I_{r+} := I_{r+1} \cup I_r$, we have

$$|\bar{a}_{r+1}(v) - \bar{a}_r(v)| = |\bar{a}_{r+1}(v - c) - \bar{a}_r(v - c)|$$
$$\leq h_{r+1}^{-1}|\langle v - c, \bar{\phi}_{r+1}\rangle_{I_{r+1}}| + h_r^{-1}|\langle v - c, \bar{\phi}_r\rangle_{I_r}|.$$

With standard direct approximation inequalities ((41) in the supplementary material), we estimate the second term as before:

$$
\begin{aligned}
h_r^{-1}|\langle v - c, \bar{\phi}_r \rangle_{I_r}| &\leq h_r^{-1}\|v - c\|_{L_p(I_r)}\|\acute{\phi}_r - \grave{\phi}_r\|_{L_q(I_r)} \\
&\leq h_r^{-1}\|v - c\|_{L_p(I_{r+})} h_r^{\frac{1}{q}} \\
&\leq h_r^{-1} h_{r+} \|v'\|_{L_p(I_{r+})} h_r^{\frac{1}{q}} \\
&\leq h_{r+} h_r^{-\frac{1}{p}} \|v'\|_{L_p(I_{r+})}.
\end{aligned}
$$

With an analogous argument on the interval $I_{r+1}$, we obtain

$$
|\bar{a}_{r+1}(v) - \bar{a}_r(v)| \leq h_{r+}\left(h_{r+1}^{-\frac{1}{p}} + h_r^{-\frac{1}{p}}\right)\|v'\|_{L_p(I_{r+})}.
$$

4. By Lemma A.5 the function $\bar{\phi}_r = \acute{\phi}_r + \grave{\phi}_r$ is non-negative. Therefore, by the mean value theorem for some $\xi \in I_r$

$$
\bar{a}_r(v) = h_k^{-1}\langle v, \bar{\phi}_r \rangle_{I_r} = h_k^{-1}v(\xi)\langle 1, \bar{\phi}_r \rangle_{I_r} \geq h_k^{-1}\min_{x \in I_r} v(x)\langle 1, \bar{\phi}_r \rangle_{I_r} = \min_{x \in I_r} v(x)\bar{a}_r(1) = \frac{1}{6}\min_{x \in I_r} v(x),
$$

where in the last step we have used Lemma A.15. Analogously, one can show that $\acute{a}_r(v) \geq \frac{1}{12}\min_{x \in I_r} v(x)$ and $\grave{a}_r(v) \geq \frac{1}{12}\min_{x \in I_r} v(x)$, using that $\acute{a}_r(1) = \frac{1}{12}$ by normalization Lemma A.6.

$\square$

**Lemma A.19.** *Let $b_r$, $r \in [\bar{m}]$ be cleaned critical breakpoints* (11), (12). *Then*

$$
|h_{r+1}^2 - h_r^2| \leq \frac{12}{\min_{x \in I_{r+}} |f''(x)|}h_{r+}\left(h_{r+1}^{2-\frac{1}{p}} + h_r^{2-\frac{1}{p}}\right)\|f'''\|_{L_p(I_{r+})}.
$$

*Proof.* From Lemma A.3 we have $h_r\langle f'', \acute{\phi}_r \rangle_{I_r} = h_{r+1}\langle f'', \grave{\phi}_{r+1} \rangle_{I_{r+1}}$ or equivalently $h_r^2 \acute{a}_r = h_{r+1}^2 \grave{a}_{r+1}$ and therefore $h_{r+1}^2 = h_r^2 \acute{a}_r / \grave{a}_{r+1}$. This implies

$$
h_{r+1}^2 - h_r^2 = \left[\frac{\acute{a}_r}{\grave{a}_{r+1}} - 1\right]h_r^2 = -\frac{1}{\grave{a}_{r+1}}[\grave{a}_{r+1} - \acute{a}_r]h_r^2.
$$

We first bound $[\grave{a}_{r+1} - \acute{a}_r]$. To this end, note that

$$
\grave{a}_{r+1}(1) - \acute{a}_r(1) = h_{r+1}^{-1}\langle 1, \grave{\phi}_{r+1} \rangle_{I_{r+1}} - h_r^{-1}\langle 1, \acute{\phi}_r \rangle_{I_r} = \frac{1}{12} - \frac{1}{12} = 0,
$$

where in the second but last step we have used the normalization properties Lemma A.6. Hence, we obtain

$$
|\grave{a}_{r+1} - \acute{a}_r| \leq h_{r+}\left(h_{r+1}^{-\frac{1}{p}} + h_r^{-\frac{1}{p}}\right)\|f'''\|_{L_p(I_{r+})},
$$

with a proof that is identical to the same bound for $|\bar{a}_{r+1} - \bar{a}_r|$ in Lemma A.18. Next, we bound

$$
\frac{1}{\grave{a}_{r+1}} \leq \frac{12}{\min_{x \in I_{r+1}} |f''(x)|}
$$

by Lemma A.18. We conclude that

$$
|h_{r+1}^2 - h_r^2| \leq \frac{12}{\min_{x \in I_{r+1}} |f''(x)|}h_{r+}h_r^2\left(h_{r+1}^{-\frac{1}{p}} + h_r^{-\frac{1}{p}}\right)\|f'''\|_{L_p(I_{r+})},
$$

Starting with $h_r^2 = h_{r+1}^2 \grave{a}_{r+1}/\acute{a}_r$ instead of $h_{r+1}^2 = h_r^2 \acute{a}_r/\grave{a}_{r+1}$ at the beginning of the proof, we obtain the same inequality with the term $h_r^2$ replaced by $h_{r+1}^2$. Thus, we can simplify to

$$|h_{r+1}^2 - h_r^2| \leq \frac{12}{\min_{x \in I_{r+}} |f''(x)|} h_{r+} \left( h_{r+1}^{2-\frac{1}{p}} + h_r^{2-\frac{1}{p}} \right) \|f'''\|_{L_p(I_{r+})},$$

which completes the proof.

$\square$

**Lemma A.20.** *Let $b_r$, $r \in [\bar{m}]$ be cleaned critical breakpoints* (11), (12). *Assume that*

$$h_{r+}^{1-\frac{1}{p}} \|f'''\|_{L_p(I_{r+})} \leq C \min_{x \in I_{r+}} |f''(x)|$$

*for a sufficiently small constant $C$ independent of $f$ and $h_r$. Then*

$$h_r \sim h_{r+1}, \qquad\qquad \bar{a}_r \sim \bar{a}_{r+1}, \qquad\qquad \mu_r \sim \mu_{r+1}.$$

*Proof.* All equivalences in this lemma are based on the following observation: For two numbers $a, b \in \mathbb{R}$ we have

$$|a - b| \leq \frac{1}{2} \max\{|a|, |b|\} \qquad\qquad \Rightarrow \qquad\qquad \frac{1}{2}a \leq b \leq 2a. \qquad (36)$$

First note that $a$ and $b$ have same sign. Indeed, if $a \geq 0$ and $b \leq 0$, we have

$$|a| + |b| \leq \frac{1}{2} \max\{|a|, |b|\} \leq \frac{1}{2}(|a| + |b|),$$

which implies $a = b = 0$. Thus, without loss of generality assume $a, b \geq 0$. In case $\min\{a, b\} = a$, we have

$$b = |b| \leq |a| + |b - a| \leq a + \frac{1}{2}a \leq \frac{3}{2}a,$$
$$b = |b| \geq |a| - |b - a| \geq a - \frac{1}{2}a \geq \frac{1}{2}a$$

and thus $\frac{1}{2}a \leq b \leq \frac{3}{2}a$. In case $\min\{a, b\} = b$, analogously we have $\frac{1}{2}b \leq a \leq \frac{3}{2}b$. Rearranging this is equivalent to $\frac{2}{3}a \leq b \leq 2a$. Using the worst of the two cases yields the claim.

We now turn to the statements of the lemma.

1. By Lemma A.19 and the given assumptions, we have

$$|h_{r+1}^2 - h_r^2| \leq c \frac{12}{\min_{x \in I_{r+}} |f''(x)|} h_{r+} \left( h_{r+1}^{2-\frac{1}{p}} + h_r^{2-\frac{1}{p}} \right) \|f'''\|_{L_p(I_{r+})}.$$
$$\leq c \frac{12}{\min_{x \in I_{r+}} |f''(x)|} h_{r+}^2 h_{r+}^{1-\frac{1}{p}} \|f'''\|_{L_p(I_{r+})}$$
$$\leq cCh_{r+}^2 \leq \frac{1}{2} \max\{h_r, h_{r+1}\} h_{r+}.$$

for sufficiently small constant $C$. It follows that

$$|h_{r+1} - h_r| = \left| \frac{h_{r+1}^2 - h_r^2}{h_{r+1} + h_r} \right| \leq \frac{1}{2} \max\{h_r, h_{r+1}\}$$

and thus with (36) we obtain $\frac{1}{2}h_r \leq h_{r+1} \leq h_r$.

2. By the first part of the lemma we have $h_r \sim h_{r+1}$ and therefore $h_r^{-1/p} + h_{r+1}^{-1/p} \lesssim h_{r+}^{-1/p}$. Thus, by Lemma A.18 and the given assumptions we have

$$|\bar{a}_{r+1} - \bar{a}_r| \le ch_{r+}\left(h_{r+1}^{-\frac{1}{p}} + h_r^{-\frac{1}{p}}\right)\|f'''\|_{L_p(I_{r+})} \le ch_{r+}^{1-\frac{1}{p}}\|f'''\|_{L_p(I_{r+})} \le cC \min_{x \in I_{r+}}|f''(x)| \le \frac{1}{2}\bar{a}_r.$$

for sufficiently small constant $C$. With (36) this implies $\frac{1}{2}\bar{a}_r \le \bar{a}_{r+1} \le \bar{a}_r$.

3. Since $\mu_r = |\bar{a}_r|^\alpha$, the previous part of the lemma directly implies $\mu_r \sim \mu_{r+1}$.

$\square$

**Lemma A.21.** *Let $P \in \mathbb{P}^{s-1}$, $1 \le s \in \mathbb{N}$ be the best linear approximation of $f''$ in $L_p$ on some interval $J \supset I_r$ and $\bar{a}_r$, $\tilde{a}_r$ be defined in 28, 30. Then for $1 \le p \le \infty$*

$$|\bar{a}_r - \bar{a}_r(P)| \lesssim |J|^s h_r^{-\frac{1}{p}}\|f^{(s+2)}\|_{L_p(J)}, \qquad\qquad |\tilde{a}_r - \tilde{a}_r(P)| \lesssim |J|^s h_r^{-\frac{1}{p}}\|f^{(s+2)}\|_{L_p(J)}.$$

*Proof.* Let $1/p + 1/q = 1$ so that $-1 + 1/q = -1/p$. Then

$$\begin{aligned}
|\bar{a}_r - \bar{a}_r(P)| = |\bar{a}_r(f'' - P)| &= h_r^{-1}\langle f'' - P, \acute{\phi}_r + \grave{\phi}_r\rangle_{I_r} \\
&\le h_r^{-1}\|f'' - P\|_{L_p(J)}\|\acute{\phi}_r - \grave{\phi}_r\|_{L_q(I_r)} \lesssim h_r^{-1}|J|^s\|f^{(s+2)}\|_{L_p(J)}h_r^{\frac{1}{q}} \\
&\le |J|^s h_r^{-\frac{1}{p}}\|f^{(s+2)}\|_{L_p(J)}.
\end{aligned}$$

The result for $\tilde{a}_r$ follows analogously.

$\square$

**Lemma A.22.** *Let $\bar{a}_r(v) \ge 0$ and for some $\beta \in \mathbb{R}$ assume*

$$h_r^\beta\|v'\|_{L_p(I_r)} \le C \min_{x \in I_{r+}}|v(x)|.$$

*Then*

$$\min_{x \in I_r}|v(x)| \le \bar{a}_r.$$

*Proof.* First assume that $\min_{x \in I_{r+}}|v(x)| = 0$. With the given assumption, this implies $v'(x) = 0$ on $I_r$ so that $v$ is constant and thus zero everywhere. In this case we have $\bar{a}_r(v) = 0$ and the result follows.

If $\min_{x \in I_{r+}}|v(x)| \ne 0$, the function $v(x)$ does not change sign and since $\bar{a}_r(v) \ge 0$, we must have $v(x) \ge 0$ for all $x \in I_r$ because $\bar{\phi}_r \ge 0$ (Lemma A.5). Then the result follows directly from Lemma A.18.

$\square$

### A.3.5 Bounds for $(I) - (IV)$ in Lemma A.12

In this section, we bound the terms (I)-(IV) in Lemma A.12. The bounds are of the form $(I) - (IV) \le [h_r^2\bar{a}_r\mu_r]h_r =: z_r h_r$, all with one extra factor $h_r$, which allows us to control the cumulative error for equidistribution over longer distances by Lemma A.13.

**Lemma A.23.** *Let $b_r$, $r \in [\bar{m}]$ be cleaned critical breakpoints (11), (12) and $\bar{a}_r, \bar{a}_{r+1} \ge 0$. Assume that*

$$h_{r+}^{1-\frac{1}{p}}\|f^{(4)}\|_{L_p(I_{r+})} \le C \min_{x \in I_{r+}}|f''(x)|$$

*for some constant $C > 0$ independent of $f$ and $h_r$. Then*

$$(I) = h_r^2[\tilde{a}_{r+1} + \tilde{a}_k + \alpha[\bar{a}_{r+1} - \bar{a}_r]]\mu_{r+1} \lesssim C[h_r^2\bar{a}_r\mu_r]h_r + \frac{1}{8}[h_{r+1}^2\bar{a}_{r+1}\mu_{r+1}]h_{r+1}.$$

*Proof.* First note that by Lemma A.16 for all $P \in \mathbb{P}^1$ we have

$$\tilde{a}_{r+1}(P) + \tilde{a}_r(P) + \alpha[\bar{a}_{r+1}(P) - \bar{a}_r(P)] = 0.$$

Hence, in case $\min_{x \in I_{r+}} |f''(x)| = 0$ the given assumption implies $f^{(4)} = 0$ so that $f'' \in \mathbb{P}^1$ and $(I) = 0$. In case $\min_{x \in I_{r+}} |f''(x)| \neq 0$, it follows that with $\bar{a}_r = \bar{a}_r(f'')$, etc.,

$$
\begin{aligned}
(I) &= h_r^2[\tilde{a}_{r+1} + \tilde{a}_k + \alpha[\bar{a}_{r+1} - \bar{a}_r]]\mu_{r+1} \\
&\quad - h_r^2[\tilde{a}_{r+1}(P) + \tilde{a}_k(P) + \alpha[\bar{a}_{r+1}(P) - \bar{a}_r(P)]]\mu_{r+1} \\
&= h_r^2[\tilde{a}_{r+1}(f'' - P) + \tilde{a}_k(f'' - P) + \alpha[\bar{a}_{r+1}(f'' - P) - \bar{a}_r(f'' - P)]]\mu_{r+1}.
\end{aligned}
\tag{37}
$$

We choose the best $L_p(I_{r+})$ approximation for $P$ and estimate all terms separately. First, we have

$$
\begin{aligned}
h_k^2\bar{a}_k(f'' - P)\mu_k = h_k^2[\bar{a}_k - \bar{a}_k(P)]\mu_k &\lesssim h_{r+}^2 h_r^{2-\frac{1}{p}}\|f^{(4)}\|_{L_p(I_{r+})}\mu_k \\
&\lesssim Ch_{r+}^2 h_r \min_{x \in I_{r+}} |f''(x)|\mu_r \lesssim Ch_{r+}^2 h_r\bar{a}_r\mu_r = C(h_r^2\bar{a}_r\mu_r)h_r,
\end{aligned}
$$

where in the second step we have used Lemma A.21, in the third our assumptions, in the fourth $|f''(x)| \lesssim \bar{a}_r$ analogous to Lemma A.22 with $\min_{x \in I_{r+}} |f''(x)| \neq 0$ and in the second but last $h_r \sim h_{r+1}$ by Lemma A.20. Analogously, we obtain

$$h_k^2\tilde{a}_k(f'' - P)\mu_k \lesssim C(h_r^2\bar{a}_r\mu_r)h_r,$$

as well as for all other combination of indices $r$ and $r+1$ because $\mu_r \sim \mu_{r+1}$ by Lemma A.20. Using these estimates for all four terms in (37), we obtain

$$(I) \lesssim C\left[(h_r^2\bar{a}_r\mu_r)h_r + (h_{r+1}^2\bar{a}_{r+1}\mu_{r+1})h_{r+1}\right],$$

which shows the lemma.

$\square$

**Lemma A.24.** *Let $b_r$, $r \in [\bar{m}]$ be cleaned critical breakpoints (11), (12) and $\bar{a}_{r+1} \geq 0$. Assume*

$$h_{r+}^{\frac{1}{2}-\frac{1}{p}}\|f'''\|_{L_p(I_{r+})} \leq C \min_{x \in I_{r+}} |f''(x)|$$

*for some constant $C > 0$ independent of $f$ and $h_r$. Then*

$$(II) = [h_{r+1}^2 - h_r^2]\tilde{a}_{r+1}\mu_{r+1} \lesssim C[h_{r+1}^2\bar{a}_{r+1}\mu_{r+1}]h_{r+1}.$$

*Proof.* By Lemma A.19 and the given assumptions, we have

$$|h_{r+1}^2 - h_r^2| \lesssim \frac{12}{\min_{x \in I_{r+}} |f''(x)|}h_{r+}\left(h_{r+1}^{2-\frac{1}{p}} + h_r^{2-\frac{1}{p}}\right)\|f'''\|_{L_p(I_{r+})} \lesssim Ch_{r+1}^{\frac{5}{2}},$$

where in the last step we have used $h_{r+1} \sim h_r$ by Lemma A.20. From Lemma A.18, the given assumptions, and $|f''(x)| \leq \bar{a}_{r+1}$ (Lemma A.22), we have

$$|\tilde{a}_{r+1}| \lesssim h_{r+1}^{1-\frac{1}{p}}\|f'''\|_{L_p(I_{r+1})} \lesssim Ch_{r+1}^{\frac{1}{2}} \min_{x \in I_{r+}} |f''(x)| \lesssim Ch_{r+1}^{\frac{1}{2}}\bar{a}_{r+1}.$$

Combining these two inequalities yields the lemma.

$\square$

**Lemma A.25.** *Let $b_r$, $r \in [\bar{m}]$ be cleaned critical breakpoints (11), (12) and $\bar{a}_r \geq 0$. Assume*

$$h_{r+}^{\frac{1}{2}-\frac{1}{p}}\|f'''\|_{L_p(I_{r+})} \leq C \min_{x \in I_{r+}} |f''(x)|$$

*for some constant $0 < C \leq 1$ independent of $f$ and $h_r$. Then*

$$(III) = \alpha^2 h_r^2\bar{a}_r^{\alpha-1}[\bar{a}_{r+1} - \bar{a}_r]^2 \lesssim C[h_r^2\bar{a}_r\mu_r]h_r.$$

*Proof.* From Lemma A.18, the given assumptions, $h_{r+1} \sim h_r$ (Lemma A.20) and $|f''(x)] \leq \bar{a}_r$ (Lemma A.22), we have

$$|\bar{a}_{r+1} - \bar{a}_r| \lesssim h_{r+} \left( h_{r+1}^{-\frac{1}{p}} + h_r^{-\frac{1}{p}} \right) \|f'''\|_{L_p(I_{r+})} \lesssim C \min_{x \in I_{r+}} |f''(x)| \lesssim C h_r^{\frac{1}{2}} \bar{a}_r.$$

Thus, with $\mu_r = \bar{a}_r^\alpha$, we have

$$(III) = \alpha^2 h_r^2 \bar{a}_r^{\alpha-1} [\bar{a}_{r+1} - \bar{a}_r]^2 \lesssim \alpha^2 h_r^2 \bar{a}_r^{\alpha-1} C^2 h_r \bar{a}_r^2 \lesssim C h_r^2 \bar{a}_r \bar{a}_r^\alpha h_r = C(h_r^2 \bar{a}_r \mu_r) h_r.$$

This completes the proof. $\qquad\square$

**Lemma A.26.** *Let $b_r$, $r \in [\bar{m}]$ be cleaned critical breakpoints (11), (12) and $\bar{a}_r, \bar{a}_{r+1} \geq 0$. Assume*

$$h_{r+}^{\frac{1}{2} - \frac{1}{p}} \|f'''\|_{L_p(I_{r+})} \leq C \min_{x \in I_{r+}} |f''(x)|$$

*for some constant $0 \leq C \leq 1$ independent of $f$ and $h_r$. Then*

$$(IV) = \alpha h_r^2 [\bar{a}_{r+1} - \bar{a}_r] R_r + h_r^2 \bar{a}_r R_r \lesssim C(h_r \bar{a}_r \mu_r) h_r.$$

*with $R_r$ defined in Lemma A.12.*

*Proof.* Recall that $R_r$ is defined by

$$R_r := \alpha(\alpha - 1)[\xi_r \bar{a}_r + (1 - \xi_r)\bar{a}_{r+1}]^{\alpha-2} [\bar{a}_{r+1} - \bar{a}_r]^2$$

for some $0 \leq \xi_r \leq 1$. From Lemma A.18, the given assumptions, $h_{r+1} \sim h_r$ (Lemma A.20) and $|f''(x)] \leq \bar{a}_r$ (Lemma A.22), we have

$$|\bar{a}_{r+1} - \bar{a}_r| \lesssim h_{r+} \left( h_{r+1}^{-\frac{1}{p}} + h_r^{-\frac{1}{p}} \right) \|f'''\|_{L_p(I_{r+})} \lesssim C h_r^{\frac{1}{2}} \min_{x \in I_{r+}} |f''(x)| \lesssim C h_r^{\frac{1}{2}} \bar{a}_r.$$

From Lemma A.20 we have $\bar{a}_r \sim \bar{a}_{r+1}$ and thus

$$\xi_r \bar{a}_r + (1 - \xi_r)\bar{a}_{r+1} \sim \bar{a}_r.$$

Combining these estimates, with $\mu_r = \bar{a}_r^\alpha$, we obtain

$$h_r^2 \bar{a}_r R_r \lesssim h_r^2 \bar{a}_r \bar{a}_r^{\alpha-2} C^2 h_r \bar{a}_r^2 \lesssim C^2 [h_r^2 \bar{a}_r \mu_r] h_r.$$

as well as

$$h_r^2 [\bar{a}_{r+1} - \bar{a}_r] R_r \lesssim h_r^2 C h_r^{\frac{1}{2}} \bar{a}_r \bar{a}_r^{\alpha-2} C^2 h_r \bar{a}_r^2 \lesssim C^3 [h_r^2 \bar{a}_r \mu_r] h_r^{\frac{3}{2}},$$

Since $C \leq 1$, we have $C^2, C^3 \leq C$, which proves the lemma.

$\qquad\square$

### A.3.6  Norm Equivalences

This section contains the equivalences of $h_r^2 \bar{a}_r \mu_r$ and $L_p$ norms.

**Lemma A.27.** *Let $1 < p \leq \infty$ and $0 < q \leq \infty$. Let $P \in \mathbb{P}^0$ be the $L_p(I_r)$ best constant approximation of some function $g$ and assume*

$$h_{r+}^{1 - \frac{1}{p}} \|g'\|_{L_p(I_{r+})} \leq C \min_{x \in I_{r+}} |g(x)|$$

*for some sufficiently small constant $C > 0$ independent of $f$ and $h_r$. Then*

 1. *$\bar{a}_r(g) \sim \bar{a}_r(P)$.*

2. $\|g\|_{L_q(I_r)} \sim \|P\|_{L_q(I_r)}$.

3. $\|g\|_{L_q(I_r)} \sim h_r^{\frac{1}{q}} |\bar{a}_r(g)|$.

*Proof.* Recall from the proof of Lemma A.20 that for two numbers $a, b \in \mathbb{R}$ we have

$$|a - b| \leq \frac{1}{2} \max\{a, b\} \qquad \Rightarrow \qquad a \sim b. \qquad (38)$$

1. In case $\bar{a}_r \geq 0$, by Lemma A.21 and $|g(x)| \leq \bar{a}_r$ (Lemma A.22) we have

$$|\bar{a}_r(g) - \bar{a}_r(P)| \leq h_r^{1 - \frac{1}{p}} \|g'\|_{L_p(I_r)} \lesssim C \min_{x \in I_{r+}} |g(x)| \lesssim C\bar{a}_r$$

   For sufficiently small $C$, with (38) this implies $\bar{a}_r(g) \sim \bar{a}_r(P)$. The case $\bar{a}_r(g) \leq 0$ follows by replacing $g$ with $-g$.

2. We first consider the case $1 \leq q < \infty$. Using direct approximation inequalities ((41) in the supplementary material), we have

$$\left| \|g\|_{L_q(I_r)} - \|P\|_{L_q(I_r)} \right| \leq \|g - P\|_{L_q(I_r)} \lesssim h_r^{1 + \frac{1}{q} - \frac{1}{p}} \|g'\|_{L_p(I_r)}$$

$$\leq Ch_r^{\frac{1}{q}} \min_{x \in I_{r+}} |g(x)| \leq C \left[ \int_{I_r} |g(x)|^q \, dx \right]^{\frac{1}{q}} \leq C\|g\|_{L_q(I_r)},$$

   which implies $\|g\|_{L_q(I_r)} \sim \|P\|_{L_q(I_r)}$ with (38) for sufficiently small $C$.

   In case $q < 1$, by Taylor's theorem, we have $u^q - v^q = q(\xi u + (1 - \xi)v)^{q-1}[u - v]$ for some $0 \leq \xi \leq 1$. With $u = |g(x)|$ and $v = |P|$, this implies

$$\|g\|_{L_q(I_r)}^q - \|P\|_{L_q(I_r)}^q = \int_{I_r} |g(x)|^q - |P(x)|^q \, dx$$

$$= \int_{I_r} q[\xi(x)|g(x)| - (1 - \xi(x))|P(x)|]^{q-1}[|g(x)| - |P(x)|] \, dx$$

$$\lesssim \min_{x \in I_r} |g(x)|^{q-1} \|g - P\|_{L_1(I_r)}$$

$$\leq \min_{x \in I_r} |g(x)|^{q-1} h_r^{1 - \frac{1}{p}} \|g - P\|_{L_p(I_r)}$$

$$\leq \min_{x \in I_r} |g(x)|^{q-1} h_r^{2 - \frac{1}{p}} \|g'\|_{L_p(I_r)}$$

$$\leq C \min_{x \in I_r} |g(x)|^{q-1} h_r \min_{x \in I_r} |g(x)|$$

$$\leq Ch_r \min_{x \in I_r} |g(x)|^q$$

$$\leq C \int_{I_r} |g(x)|^q \, dx$$

$$\leq C\|g(x)\|_{L_q(I_r)}^q,$$

   where in the third step we have used $q - 1 < 0$ and that $P = g(\eta) \Rightarrow |P| \geq \min_{x \in I_r} |g(x)|$ for some $\eta \in I_r$, as can easily be seen by by first order optimality criteria and the mean value theorem. In the fourth step we have used that $\|\cdot\|_{L_1(I_r)} \leq h_r^{1 - \frac{1}{p}} \|\cdot\|_{L_p(I_r)}$ by Hölder's inequality and in the sixth the given assumptions. Again with (38) this implies $\|g\|_{L_q(I_r)}^q \sim \|P\|_{L_q(I_r)}^q$ for sufficiently small $C$ and thus the statement of the lemma by taking the $q$-th root.

3. We first show the desired identities for $P$ instead of $g$. Indeed, we have

$$\|P\|_{L_q(I_r)} = h_r^{\frac{1}{q}}|P|,$$

$$h_r^{\frac{1}{q}}\bar{a}_r(P) = h_r^{\frac{1}{q}}[h_r^{-1}\langle P, \bar{\phi}_r\rangle_{I_r}] = \frac{1}{12}h_r^{\frac{1}{q}}P.$$

With the first two equivalences of this lemma, this implies

$$\|g\|_{L_q(I_r)} \sim \|P\|_{L_q(I_r)} = h_r^{\frac{1}{q}}|P| \sim h_r^{\frac{1}{q}}|\bar{a}_r(P)| \sim h_r^{\frac{1}{q}}|\bar{a}_r(g)|,$$

which concludes the proof.

$\square$

## A.4 Approximation

In this section, we prove the main approximation result, restated here for convenience:

**Theorem A.28** (Theorem 3.4, restated). *Let $\theta$ be a critical point (11), with cleaned breakpoints in ascending order 12. For $r, s \in \{2, \ldots, \bar{m}\}$, let $\mathcal{I} = \{r, r+1, \ldots, s\}$ be a set of consecutive neurons with $D_{\mathcal{I}} := \bigcup_{k \in \mathcal{I}} I_k$ and*

$$\max\left\{ h_{k+}^{\frac{1}{2}-\frac{1}{p}}\|f^{(3)}\|_{L_p(I_{k+})},\ h_{k+}^{1-\frac{1}{q}}\|f^{(4)}\|_{L_q(I_{k+})},\ \right\} \le C \min_{x \in I_{k+}} |f''(x)| \tag{39}$$

*for some $1 < q, p \le \infty$ and some sufficiently small constant $C > 0$ independent of $f$ and $h_k$. Then*

$$\|f_\theta - f\|_{L_2(D_{\mathcal{I}})} \lesssim |\mathcal{I}|^{-2}\|f''\|_{L_{2/5}(D_{\mathcal{I}})}. \tag{40}$$

Since we have already established equidistribution in Theorem 3.3, the approximation results is standard.

*Proof.* We first split the $L_2$ norm

$$\|f_\theta - f\|_{L_2(D_{\mathcal{I}})}^2 = \sum_{r \in \mathcal{I}} \|f_\theta - f\|_{L_2(I_r)}^2$$

$$= \sum_{r \in \mathcal{I}} \left[\|f_\theta - f\|_{L_2(I_r)}^{\frac{2}{5}}\right]^5$$

$$= \sum_{r \in \mathcal{I}} \left[\frac{1}{|\mathcal{I}|}\sum_{s \in \mathcal{I}} \|f_\theta - f\|_{L_2(I_r)}^{\frac{2}{5}}\right]^5,$$

where in the last step we have inserted an artificial sum for later use. By (22) and the discussion thereafter on each interval $I_r$ the neural network is a best linear approximation and therefore by standard direct approximation results ((41) in the supplementary material), we have

$$\|f_\theta - f\|_{L_2(D_{\mathcal{I}})}^2 \lesssim \sum_{r \in \mathcal{I}} \left[\frac{1}{|\mathcal{I}|}\sum_{s \in \mathcal{I}} h_r^{\frac{2}{5}\left(2+\frac{1}{2}-1\right)}\|f''\|_{L_1(I_r)}^{\frac{2}{5}}\right]^5.$$

By Lemma A.27, we have

$$h^{\frac{3}{5}}\|f\|_{L_1(I_r)}^{\frac{2}{5}} \sim h^{\frac{3}{5}}[h_r|\bar{a}_r|]^{\frac{2}{5}} = [h_r^{\frac{5}{2}}|\bar{a}_s|]^{\frac{2}{5}} \sim \|f''\|_{L_{2/5}(I_r)}^{\frac{2}{5}}$$

and therefore

$$\|f_\theta - f\|_{L_2(D_{\mathcal{I}})}^2 \lesssim \sum_{r \in \mathcal{I}} \left[\frac{1}{|\mathcal{I}|}\sum_{s \in \mathcal{I}} \|f''\|_{L_{2/5}(I_r)}^{\frac{2}{5}}\right]^5.$$

As a side remark, we could have used $\|f_\theta - f\|_{L_2(I_r)} \lesssim \|f\|_{B^2_{2/5}(L_{2/5}(I_r))}$ directly if we would use Besov norms. Anyways, note that the sum depends on $s$, but the summands depend on $r$, which we fix with equidistribution $\|f''\|_{L_{2/5}(I_r)} \sim \|f''\|_{L_{2/5}(I_s)}$ from Theorem 3.3. Then

$$
\begin{aligned}
\|f_\theta - f\|^2_{L_2(D_\mathcal{I})} &\lesssim \sum_{r \in \mathcal{I}} \left[ \frac{1}{|\mathcal{I}|} \sum_{s \in \mathcal{I}} \|f''\|^{\frac{2}{5}}_{L_{2/5}(I_s)} \right]^5 \\
&= \sum_{r \in \mathcal{I}} \left[ \frac{1}{|\mathcal{I}|} \|f''\|^{\frac{2}{5}}_{L_{2/5}(D_\mathcal{I})} \right]^5 \\
&= \frac{1}{|\mathcal{I}|^4} \|f''\|^2_{L_{2/5}(D_\mathcal{I})},
\end{aligned}
$$

which concludes the proof.

$\square$

# B  Technical Supplements

## B.1  Besov Spaces

For integer $s \geq 0$ and $1 \leq p \leq \infty$, Sobolev norms are defined by

$$
\|f\|^p_{W^{s,p}(D)} := \sum_{r=0}^{s} |f|^p_{W^{s,p}(D)}, \qquad\qquad |f|_{W^{s,p}(D)} := \|f^{(r)}\|_{L_2(D)}
$$

For Besov norms, define the difference operators $(\Delta^1_h f)(x) := f(x+h) - f(x)$ and $\Delta^r_h := \Delta^1_h \Delta^{r-1}_h$, extended by zero in case $x + h \notin D$, and the $r$-th order modulus of smoothness

$$
\omega_r(f, t)_p := \sup_{|h| \leq t} \|\Delta^r_h f\|_{L_p(D)}.
$$

Then for $0 < p, q < \infty$, and the smallest integer $r > s$, the Besov norms are defined by

$$
\|f\|_{B^s_q(L_p(\Omega))} := \|f\|_{L_p(D)} + |f|_{B^s_q(L_p(\Omega))}, \qquad |f|_{B^s_q(L_p(\Omega))} := \left\{ \int_0^\infty \left[ t^{-s} \omega_r(f, t)_p \right]^q \frac{dt}{t} \right\}^{\frac{1}{q}}.
$$

See DeVore & Lorentz (1993); DeVore (1998) for details.

## B.2  Direct Approximation Estimates

For the best $L_p$ approximation with polynomials $\mathbb{P}^{r-1}$ of degree at most $r - 1$ on interval $I$ it is well known that

$$
\inf_{p \in \mathbb{P}^{r-1}} \|f - p\|_{L_p(I)} \lesssim |I|^{r + \frac{1}{p} - \frac{1}{q}} \|f^{(r)}\|_{L_q(I)} \tag{41}
$$

for all $r > 0$ and $1 \leq p, q \leq \infty$ with $r + \frac{1}{p} - \frac{1}{q} > 0$. See e.g. DeVore (1998), (6.9).

## B.3  Main Results with Besov Norms

In the main Theorem 3.4 we use the Sobolev type norm $\|f''\|_{L_q(D_\mathcal{I})}$, which is unusual for $q := 2/5 < 1$. This is permissible, because the assumptions (13) requires higher weak derivatives in regular $L_p$ norms with $1 \leq p \leq \infty$. In this section, we consider a similar result in Besov norms. These allow a larger range of $q, p < 1$ in the assumptions. Up to an arbitrarily small discrepancy in smoothness, the approximation bounds use the same norms than classical adaptive approximation in (5).

**Theorem B.1.** *Let $\theta$ be a critical point (11), with cleaned breakpoints in ascending order 12. For $r, s \in \{2, \ldots, \bar{m}\}$, let $\mathcal{I} = \{r, r+1, \ldots, s\}$ be a set of consecutive neurons with $D_{\mathcal{I}} := \bigcup_{k \in \mathcal{I}} I_k$ and assume*

$$h_k^{\frac{1}{2} - \frac{1}{o}} \|(\Delta_t^2 f)'\|_{L_o(I_{k+})} \leq C \min_{x \in I_{k+}} |(\Delta_t^2 f)(x)|, \tag{42}$$

$$h_{k+}^{\frac{1}{2} - \frac{1}{p}} |f''|_{B_p^1(L_p(I_{k+}))} \leq C \min_{x \in I_{k+}} |f''(x)| \neq 0, \tag{43}$$

$$h_{k+}^{1 - \frac{1}{q}} |f''|_{B_q^2(L_q(I_{k+}))}, \leq C \min_{x \in I_{k+}} |f''(x)| \neq 0, \tag{44}$$

*uniformly for all $t > 0$, $o = 2/5$, some $1 \leq o \leq \infty$, some $\frac{1}{2} \leq p \leq \infty$, some $\frac{1}{3} \leq q \leq \infty$ and a sufficiently small constant $C > 0$ independent of $f$ and $h_k$. Then*

$$\|f_\theta - f\|_{L_2(D_{\mathcal{I}})} \lesssim |\mathcal{I}|^{-2} |f|_{B_q^s(L_q(I_r))}$$

*for every $s < 2$.*

*Proof.* The result is proven analogously to Theorem 3.4 with a few small changes that we point out in the following.

1. *Assumptions:* Assumptions (43), (44) yield

$$\max \left\{ h_{k+}^{\frac{1}{2} - \frac{1}{p}} |f''|_{B_p^1(L_p(I_{k+}))}, h_{k+}^{1 - \frac{1}{q}} |f''|_{B_q^2(L_q(I_{k+}))}, \right\} \leq C \min_{x \in I_{k+}} |f''(x)| \neq 0,$$

   analogous to 13 with Sobolev norms replaced by Besov norms, which allow the larger ranges $\frac{1}{2} \leq p \leq \infty$ and $\frac{1}{3} \leq q \leq \infty$. Reconsidering the proof of Theorem 3.4, the non-zero condition on the left hand side ensures the second case in the proof of Lemma A.22. Then, we replace the use of the approximation inequality (41) with

$$\inf_{p \in \mathbb{P}^{r-1}} \|f - p\|_{L_p(I)} \lesssim |I|^{r + \frac{1}{p} - \frac{1}{q}} \|f\|_{B_q^r(L_q(I))}$$

   with $r > 0$ and $r + \frac{1}{p} - \frac{1}{q} > 0$, which remains true in case $q < 1$, see e.g. DeVore (1998), (6.8). We make this replacement in the proofs of Lemmas A.18, A.19, A.21 and A.27, where we obtain minimal $p, q$ if we approximate in the $L_1$ norm after applying Hölder's inequality.

2. *Conclusion:* By assumption (42) and Lemma A.27, for any $0 < \rho < \infty$ and $s < 2$ by we have

$$h_r^{\frac{1}{q}} \omega_2(f, t)_q \sim h_r^{\frac{1}{p}} \omega_2(f, t)_p \qquad \Rightarrow \qquad h_r^{\frac{1}{q}} |f|_{B_\rho^s(L_q(\Omega))} \sim h_r^{\frac{1}{p}} |f|_{B_\rho^s(L_p(\Omega))}.$$

   For $1 \leq p \leq \infty$, it is well known that Sobolev and Besov spaces are closely related. Using this in the second step below and Hölder's inequality in the first, we have

$$\|f''\|_{L_q(I_r)} \leq h_r^{\frac{1}{q} - \frac{1}{p}} \|f''\|_{L_p(I_r)} \leq h_r^{\frac{1}{q} - \frac{1}{p}} |f|_{B_q^s(L_p(I_r))} \sim |f|_{B_q^s(L_q(I_r))}.$$

   Plugging this into the approximation bounds of Theorem 3.4, we obtain

$$\|f_\theta - f\|_{L_2(D_{\mathcal{I}})} \lesssim |\mathcal{I}|^{-2} \|f''\|_{L_{2/5}(D_{\mathcal{I}})} \lesssim |\mathcal{I}|^{-2} |f|_{B_q^s(L_q(I_r))}$$

   for any $s < 2$, which concludes the proof.

$\square$

### B.4  Networks in Inner Weights

In this section we prove Lemma 3.1. Without loss of generality, we use the domain $D = [-1, 1]$. We first show an abstract characterization of critical points, similar to the proof sketch in Section 4.

**Lemma B.2.** *For an arbitrary function $\theta \to f_\theta \in L_2(D)$, the weights $\theta \in \mathbb{R}^m$ are a critical point of the loss $\|f_\theta - f\|_{L_2(D)}^2$ if and only if*

$$\langle f_\theta - f, v \rangle = 0, \qquad\qquad v \in \mathrm{span}\{\partial_{\theta_r} f_\theta : r \in [m]\}.$$

*Proof.* The critical points are given by

$$\langle f_\theta - f, \partial_{\theta_r} f_\theta \rangle = 0, \quad r \in [m].$$

Clearly, the condition of the lemma implies the critical point condition. In the other direction, taking linear combinations of the critical point condition directly implies the condition of the lemma.

$\square$

*Proof of Lemma 3.1.* We first show that $f_{W,V,B}$ and $f_{w,b}$ represent the same functions and then relate their critical points.

1. *Construction of $f_{w,b}$:* We use the property $\sigma(ax) = a\sigma(x)$ for all $a \geq 0$ of ReLU activations to rewrite $F_{W,V,B}$ as

$$F_{W,V,B} := \left( B_0 + \sum_{V_r=0} W_r \sigma(B_r) \right) + W_0 x + \sum_{W_r, V_r \neq 0}^m W_r |V_r| \sigma\left( \mathrm{sign}(V_r)x - \frac{B_r}{|V_r|} \right).$$

   This is already in the same format as $f_{w,b}$, except for the term $\mathrm{sign}(V_r)$ inside the activation. We can easily eliminate it with the formula

$$\sigma(-x + b) = \sigma(x - b) - (x - b) \tag{45}$$

   and obtain

$$F_{W,V,B} := \left( B_0 + \sum_{V_r=0} W_r \sigma(B_r) - \sum_{V_r<0} W_r B_r \right) + \left( W_0 - \sum_{V_r<0} W_r |V_r| \right) x + \sum_{W_r, V_r \neq 0}^m W_r |V_r| \sigma\left( x - \frac{B_r}{V_r} \right).$$

   after rearranging terms. Note that the first two parenthesis are constant and thus, we can find $f_{w,b}$ by matching terms. The last formula also shows that the breakpoints are $B_r/V_r$ for all $r$ with nonzero $W_r$ and $V_r$.

2. *Critical Points:* Let $F_{W,V,B}$ be a critical point and $f_{w,b}$ be the corresponding network constructed above. We show that the latter is also a critical point for optimization of $w$ and $b$. To this end, define the linear spaces

$$X_f := \mathrm{span}\{\partial_\square f_{w,b} : \square \in \{w_r, b_r\}, \, r \in [m]\}.$$
$$X_F := \mathrm{span}\{\partial_\square f_{W,V,B} : \square \in \{W_r, V_r, B_r\}, \, r \in [m]\}.$$

   Since $f_{w,b} = F_{W,V,B}$, by Lemma B.2 it suffices to show that $X_f \subset X_F$. First note that

$$\partial_{b_0} f_{w,b} = 1 = \partial_{B_0} F_{W,V,B} \in X_F$$
$$\partial_{w_0} f_{w,b} = x = \partial_{W_0} F_{W,V,B} \in X_F.$$

   For $r > 0$, we have

$$\partial_{w_r} f_{w,b} = \sigma\left( x - \frac{B_r}{V_r} \right) \in \mathrm{span}\left\{ 1, x, \frac{1}{|V_r|}\sigma(V_r x - B_r) \right\}$$
$$= \mathrm{span}\left\{ \partial_{B_0} F_{W,V,B}, \partial_{W_0} F_{W,V,B}, \partial_{W_r} F_{W,V,B} \right\} \subset X_F,$$

where in the first span we have used (45) if $V_r < 0$ and by construction we know that $W_r, V_r \neq 0$. Analogously, we obtain

$$\partial_{b_r} f_{w,b} \;=\; w_r \dot{\sigma}\left(x - \frac{B_r}{V_r}\right) \;\in\; \text{span}\left\{1, \dot{\sigma}(V_r x - B_r)\right\} \;=\; \text{span}\left\{\partial_{B_0} F_{W,V,B}, \partial_{B_r} F_{W,V,B}\right\} \;\subset\; X_F,$$

Thus, all partial derivatives of $f_{w,b}$ are contained in $X_F$ so that $X_f \subset X_F$, which concludes the proof

$\square$

