# OpenReview forum: "Nonlinear Behaviour of Critical Points for a Simple Neural Network"
_TMLR — Accepted by TMLR_

### Review · Reviewer_YTYo · 2024-07-11

**Summary Of Contributions:**

This paper presents a detailed investigation of the nonlinearity inherent in a simple shallow network operating in one dimension. The analysis reveals that these networks can exhibit unfavorable critical points, which can be mitigated by ensuring a sufficiently high local resolution. The paper demonstrates that when the local resolution is sufficiently high, all critical points of the network satisfy L2 loss bounds that are optimal in terms of adaptive approximation in Sobolev and Besov spaces. Crucially, these bounds cannot be matched by linear approximation methods, highlighting the inherent nonlinearity and global behavior of the critical points' inner weights.

**Audience:**

Yes

**Claims And Evidence:**

Yes

**Requested Changes:**

1. Could you provide additional clarification on Assumption (13)? Any concrete example?

2. The paper is primarily theoretical, without any empirical validation. Could you provide analogous illustrations based on experimental results?

3. In what ways could your findings be utilized to enhance deep learning algorithms?

4. What are the primary challenges in generalizing the results to high dimensions?

**Strengths And Weaknesses:**

Strengths:
1. The paper conducts a detailed examination of the nonlinearity present in a basic shallow neural network, moving beyond the oversimplifications of the over-parametrized regime. This analysis delivers valuable insights into the intricate optimization landscape of real-world neural networks.
2. The paper recognizes the existence of unfavorable critical points within the neural network and demonstrates that these critical points can be alleviated by mitigated by ensuring a sufficiently high local resolution.

Weaknesses:
1. The analysis is limited to a simple shallow network in one dimension, which may not be fully representative of the complexity of practical neural network architectures.
2. While the theoretical insights are valuable, the paper does not explore the practical implications or applications of these findings.

---

> ### Author Response · Authors · 2024-08-30
> **Reply**
>
> Thank you for the constructive review. It helped us to improve the manuscript. Changes are marked in blue.
>
> Weaknesses:
>
> 1. It is true that the setup is quite simple. Nevertheless, the argument to combine critical point analysis with approximation theory is new, to the best of our knowledge. For some discussion of extensions to more realistic problems, see "Requested Changes" Items 3 and 4 below.
>
> 2. See Item 3 below.
>
> Requested Changes:
>
> 1. This is a fair point. We've added two new subsections:
>
>    - Section 5.2: Provides an informal motivation for the assumption and includes an example.
>    - Section 5.3: Discusses the feasibility of the assumption.
>
> 2. We agree that some experiments are beneficial and added Section 6 with some preliminary ones.
>
> 3. The theory needs to be further developed, especially for more realistic networks and full gradient descent dynamics, for a thorough answer to this question. As it stands now, the theory indicates that gradient descent is good at allocating the networks resources in convex and concave sub-domains, but not necessarily between them. One may therefore search for a strategy to mitigate these shortcomings. For example, a rebalancing of neurons in sub-domains by growing and pruning would be one possibility. Also deeper layers can create and remove breakpoints in the network output, so that careful architectures may help.
>
>    Extensions to higher dimensions (Item 4) may provide similar and more practical conditions when GD succeeds, when it fails and ideas how to improve it.
>
>
> 4. The general strategy seems plausible for the multi-dimensional case: The decoupling of inner and outer weights by the construction of the complement space $X^\perp$ works and we can again consider the limit of infinitely many weights. But multiple dimensions also require some differences:
>
>    - In $1d$ the space $X^\perp$ can be constructed explicitly. This seems implausible for multiple dimensions. One can try abstract characterizations or simpler approximations $X^\perp_m \approx X^\perp$ with the same limiting behaviour for wide networks.
>
>    - One also has to replace the smoothness spaces. Nonlinear approximation of shallow multi dimensional networks can be characterized by Barron, variation, or Radon-BV smoothness.
>
>    The counter example for bad critical points is easy to extend to multiple dimensions.

---

### Review · Reviewer_xGKk · 2024-08-11

**Summary Of Contributions:**

The paper studies the training behavior of a simple shallow neural network in one dimension. The over-parametrization regime of neural networks has been extensively studied via linearization and landscape analysis. However, these analysis techniques are not quite sufficient to capture the nonlinear dynamics when the network is much less over-parametrized.

**Audience:**

No

**Claims And Evidence:**

No

**Requested Changes:**

My primary concern with this submission is, in my opinion, a lack of clarity in the exposition. I found it incredibly difficult to read this paper and parse what its contribution is, both conceptually and technically. Therefore, my biggest requested change is to drastically improve the writing of this paper for it to be intelligible to readers beyond just those that work on this specific topic.

**Strengths And Weaknesses:**

I found the writing of this submission extremely unclear. Perhaps I'm not the right audience for this, but I would have greatly preferred (1) a clearer distinction between known work and the paper's contributions, (2) more background, and definitions of jargon specific to this area; (3) more details on the implications of the paper's results.

---

> ### Author Response · Authors · 2024-08-30
> **Reply**
>
> Strengths and Weaknesses:
>
> 1. The approximation properties of gradient descent trained neural networks are poorly understood. The papers in the Introduction->"Approximation" that include training require smoothness from linear approximation methods, or use alternative optimizers like greedy methods. The papers in Introduction->"Beyond Linearization" exploit nonlinearity for special target functions like ridge functions. Results that provide approximation guarantees for gradient descent trained networks, or even critical points, on full smoothness classes of nonlinear approximation are to the best of our knowledge unknown. This also justifies the simple model problem in the manuscript.
>
> 2. The manuscript contains a short introduction to approximation theory of piecewise linear functions (Section 2) and provides the Besov spaces (Appendix B.1) and approximation inequalities (Appendix B.2) it uses. Since this is a research paper, not a book or survey, it has to strike a balance between review of background material and brevity.
>
> 3. We've added a more careful discussion of assumption (13) (Sections 5.2 and 5.3) and implications for networks with inner input weights (at the beginning of Section 3). See also Items 3,4 in the reply to Reviewer YTYo for some implications and extensions. If there are any concrete questions, we are happy to address them.
>
> Requested Changes:
>
> Can you provide some concrete examples? Then we are happy to address them.

---

### Review · Reviewer_ZmLz · 2024-08-17

**Summary Of Contributions:**

The paper studies the critical points of a two-layer, 1-dim-input, 1-dim-output simplified network. Specifically, the paper shows that under certain conditions, critical points approximate the ground truth with diminishing error with order $m^{-2}$, where $m$ is the width of the network. The paper claims the following major insights:

1. The result shows that these critical points "exploit the non-linearity of the network" because linear approximation can not achieve the same result. Precisely, linear approximation can also achieve $m^{-2}$ approximation error, but only for ground truth functions in a much smaller class.

2. The results are achieved by sufficiently high resolution, where resolution is measured by the size of intervals between two consecutive bias terms $b_i$ and $b_{i+1}$. Under sufficiently high resolution, the neural network approximates every concave or convex subdomain of the ground truth with a linear segment.

3. Unfavourable critical points (with high approximation loss) can also be constructed.

**Audience:**

Yes

**Claims And Evidence:**

No

**Requested Changes:**

According to the weakness, I suggest to:

1. Improve the presentation;

2. Compare with the above references;

3. Clearly justify the contribution of this paper.

**Strengths And Weaknesses:**

### Strength

The paper provides a fine-grained analysis of the approximation ability of the studied network, with a comparison with the linear approximation. The paper also provides mathematical insights into why the approximation result can be established.

### Weakness

**Weakness 1: Lack of Clarity and Readability**: The paper is very difficult to follow. **First**, some of the terminologies and concepts appear to be directly borrowed from other fields, such as finite element analysis. which I believe are unfamiliar to the broader machine learning community. For example, on Page 3:

1. The definition of "normalized grid size" is unclear. It is related to the interval between $b_i$'s. I don't know why multiplying by the network width $m$ is called "normalization".

2. The term "local resolution" is never clearly defined. I had to guess its meaning.

3. "Generally, this requires global movement of breakpoints $b_r$ from $f$ independent initial locations". Why is the movement of $b_r$ independent of initial locations? If the initial locations are appropriate, why would they need to be moved?

4. "have highly oscillatory features between breakpoints that are imperceptible to the gradient.". The term "oscillatory feature" is not defined, and what does this whole sentence mean?

These are just a few examples from a single page; there are numerous other instances throughout the paper.

**Second**, the technical content is presented in a manner that is not accessible to general readers. Key arguments are often preceded by lengthy and complex primers, making the main points difficult to grasp. For example, in Section 2, the crucial argument "Therefore, if we can find critical points ... it must exploit the nonlinearity of the network" is buried at the end of the section.

**Weakness 2: Lack of Discussion on Important References**: The paper does not adequately cite or discuss key references. For instance, in the context of landscape studies, [R1] and [R2] have shown that bad local minima exist in arbitrarily deep and wide neural networks with smooth and piecewise linear activations. Specifically, [R2] demonstrates that arbitrarily bad local minima can exist in ReLU networks, in a more general setting than that considered in this paper.

**Weakness 3: Unclear Contribution**: I have several concerns regarding the contributions claimed by this paper:

1. The paper claims that it studies the approximation error of critical points, distinguishing itself from universal approximation studies. However, (1) it does not characterize all critical points, and (2) it only considers training error in a finite domain. In this case, a sufficiently wide network can be trained very well with $\theta$ in a sufficiently large compact set. Then analyzing the universal approximating error is equivalent to analyzing the the global minimum (a critical point) of the training problem.

2. The setting is overly simplified. The network has only one input dimension and has no input weights (only input bias is considered). This setup resembles a one-dimensional interpolation problem rather than a typical neural network. I doubt it can provide sufficient insight into general neural networks. Additionally, the authors frequently reference splines, but it is unclear whether the techniques used in this paper are direct extensions of approximation analysis methods from interpolation. The authors should clarify this point.

3. The claim that the paper analyzes "nonlinear behavior" is overstated. The paper merely shows that nonlinear models can achieve better approximation than linear models, which is not a surprising result. The paper does not study any training dynamics. In particular, the paper references that NTK is a "linearization argument", treating it as an important motivation. However, NTK studies do analyze training dynamics, as they typically focus on the convergence of training.

[R1] Tian Ding, Dawei Li, and Ruoyu Sun. "Suboptimal local minima exist for wide neural networks with smooth activations." Mathematics of Operations Research 47.4 (2022): 2784-2814.

[R2] Fengxiang He, Bohan Wang, and Dacheng Tao. "Piecewise linear activations substantially shape the loss surfaces of neural networks." ICLR 2020.

---

> ### Author Response · Authors · 2024-08-30
> **Reply**
>
> Thank you for the review. Comments are below and changes in the manuscript are marked in blue.
>
> *Weakness 1:*
>
> 1. Multiplying by $m$ is equivalent to dividing by the size $h \sim 1/m$ of a uniform grid so that the normalized grid size is the ratio between the actual $h_r$ and a uniform reference $h$. We've added some extra explanation after (4) and in Section 3, "Equidistribution".
>
> 2. This term is used colloquially in introduction and motivations, not in any rigorous sense. It refers to the local size $h_r$, which we have added at its first occurrence.
>
> 3. The sentence says that the initial location is independent of $f$, not that the movement is independent of initial location. We have reworded this slightly to make it more clear. If the initial $b_r$ are appropriate, their movement is zero and included in the general statement.
>
> 4. This is similar to Item 2. The quote is from the introduction, which is informal. A concrete example is constructed in Section 5.1. We've added a reference.
>
> Section 2 is a short introduction to approximation theory, not a primer for the quote at the end (although we agree that it is an important point). The perception seems different for other reviewers. We followed the advise from Reviewer Pdkp not to alter it.
>
> *Weakness 2:*
>
> Thank you for the references. We've included them in the literature review on landscape analysis at the beginning of the manuscript, with a short discussion of the latter.
>
> Please note that the main contribution of the manuscript is not the construction of bad local minima, but a criterion to obtain good critical points, characterized by their approximation properties. We have cited all literature that we are aware of, but to the best of our knowledge, approximation theory of generic critical points seems to be unknown.
>
> *Weakness 3:*
>
> 1. "[...] analyzing the universal approximating error is equivalent to analyzing the the global minimum (a critical point)". The manuscript is not concerned with the approximation error at one critical point, the global minimum. It rather considers the approximation error at all critical points of sufficiently big networks. This is a very different question and not answered by standard theory.
>
> 2. A discussion of input weights has been added at the beginning of Section 3, in Lemma 3.1 and before Section 4. For potential generalizations to higher dimensions, see Item 4 for Reviewer YTYo.
>
>    The question addressed in the manuscript is to what extend a general purpose gradient descent optimizer (or rather their limit, the critical points) can achieve optimal approximation rates, without any specialized understanding of underlying problem. In contrast, algorithms in the spline and finite element literature do use such an understanding. Instead of optimizing the $L_2$ loss, they estimate the equidistribution and optimize it directly, either by refinement or mesh movement.
>
> 3. "The claim that the paper analyzes "nonlinear behavior" is overstated. The paper merely shows that nonlinear models can achieve better approximation than linear models, ...". This is well known and not the point of the manuscript. It shows that this better nonlinear approximation can be expected from any critical points with enough local breakpoints and therefore also for the outcome of gradient descent training. The mentioned NTK theory does not provide that, as discussed in "Comparison with NTK Theory" at the end of Section 3. Results for the full training dynamics are left for future work, see Reviewer Pdkp Item 4 for some information.
>
> *Requested Changes*
>
> 1. We improved the items above, but omitted the ones that conflict with other reviewers.
>
> 2. We've included the references with a short description. Note that the spurious local minima constructed in these do not contradict the main results of the manuscript. The critical points addressed in the main theorems do not need to be global minimizers. Even if they are not, they still maintain favourable approximation properties.
>
> 3. Given the discussion in "Weaknesses 3", it seems that the reviewer has misunderstood some of the key takeaways. We hope the answers can clarify some of these.

---

### Review · Reviewer_Pdkp · 2024-08-23

**Summary Of Contributions:**

Authors consider a $D=1$ approximation problem with ridge function of the form $\hat{f}(w, b) = \sum_{r=1}^{m}w_r\text{ReLU}(x - b_r),\,w_r, b_r \in \mathbb{R}$ adapted to the setting of gradient-based method applied to $L_2$ loss $L(w, b) = \left\\|\hat{f} - f\right\\|_2^2$. More specifically, they argue that gradient-based method introduce small adjustments to the parameters until they reach critical point $w^{\star}, b^{\star}$ with $\left<f -\hat{f}(w^{\star}, b^{\star}), \partial_w \hat{f}(w^{\star}, b^{\star})\right> = \left<f -\hat{f}(w^{\star}, b^{\star}), \partial_b \hat{f}(w^{\star}, b^{\star})\right> = 0$. So they study how the approximation error scales with the number of parameters at a critical point specified above. In this setting they show that for “good” critical points error scales optimally with the number of parameters. “Bad” critical points exist and may have an arbitrarily large error that is not reducible by gradient-based technique alone. A condition is given that defines a “good” critical point and an argument is made about what may constitute a “bad” critical point.

**Audience:**

Yes

**Claims And Evidence:**

Yes

**Requested Changes:**

Requested Changes

1. Authors consider $\text{ReLU}$ activation function that has very useful property $\text{ReLU}(ax) = |a|\text{ReLU}\left(\text{sign}(a)x\right)$. It seems to me that the space of functions $\hat{f}(w, b) = \sum_{s=\pm 1}\sum_{r=1}^{m}w_{r,s}\text{ReLU}(s x - b_{r,s}),\,w_{r,s}, b_{r,s} \in \mathbb{R}$ is the same as a space of $\sum_{r}w_r\text{ReLU}(a_rx + b_r)$. Given that, the results of the authors should be possible to extend in the case of standard feedforward neural networks with $\text{ReLU}$ activations and a single hidden layer. Can the authors please comment on that?
2. Conditions (13) and (14) are clearly very important. Can the authors please provide a more intuitive explanation in style of Section 5 on the meaning of these conditions. What kind of function and grid may violate this condition? What are the examples when this condition holds?
3. Discussion in Section 5 is very helpful and intuitive. Indeed, it is hard to argue that under these unfavorable conditions local adjustments of weights is futile and gradient-based methods fail. However, it is not explained clearly enough how this is related to the main result given in Theorem 3.3 and to the conditions (13) and (14). Is it possible to transfer aforementioned conditions to a practical conclusion? For example, if I have a function from a certain class is it always possible to select a fine enough grid (the choice of $b_r$) such that condition (13) / (14) holds? If this is the case, can the authors construct examples that demonstrate this property?
4. I also want to point out that the framing of the obtained result is somewhat not accurate. Authors refer to the “good” (e.g., dense enough to result in equidistribution) locations of $b_r$ as “sufficiently fine resolution.” This sounds a little misleading, since $b_r$ are selected at will only during the initialisation, unless authors want to bypass gradient descent and redistribute / add $b_r$ according to some other training rule (e.g., greedy). Can the authors please comment on that part? Is it possible to relate main result to initialisation and, when the number of $b_r$ is large to the overparameterized case (at least for sufficiently smooth targets)?

**Strengths And Weaknesses:**

I appreciate that authors provide explanations and context from approximation theory. I think these parts substantially increase a potential readership of the paper and advise against removing or altering them. Overall, I find organization of the paper to be clear enough for the reader with modest experience in approximation theory.

That said, in my opinion some explanations and the framing of the main results can be improved. This is further explained in the next section of the review.

---

> ### Author Response · Authors · 2024-08-30
> **Reply**
>
> Thank you for the constructive feedback. It was very helpful to improve the manuscript. Changes are marked in blue.
>
> Requested Changes:
>
> 1. This is a good observation. One has to be a little careful because the functions $\sigma(x-b_r)$ are all zero left of all breakpoints, while the standard from is not. This can be fixed by adding the two basis functions $1$ and $x$ to the network. One can then show that also the critical points carry over. This allows us to transfer equidistribution properties of the breakpoints from the main results to networks in standard form. We included this in the text and make the following changes:
>
>    - Rewritten the introduction of the networks at the beginning of Section 3.
>    - Added Lemma 3.1, which connects critical points of both forms of the networks. Added the proof to the appendix.
>    - Added a discussion of equidistribution for standard networks before Section 4.
>
> 2. This is a fair point, we've added
>
>    - Section 5.2: Provides an informal motivation for the assumption.
>    - Section 5.3: Discusses the feasibility of the assumption.
>
> 3. We added a more thorough discussion to the new Section 5.3. The assumption is easy to satisfy in convex/concave regions of smooth functions, given that the critical point has sufficient local resolution. Of course this can only be seen a-posteriori, once we have a critical point. The new section, as well as Item 4 below, also contain an outlook for a-priori conditions.
>
> 4. As the reviewer points out, the "sufficient resolution" refers to the $b_r$ at the critical points. To ensure a-priori (before training) that they are satisfied by the gradient descent limit, we have to consider the training dynamics. This is not done in the manuscript, but one may consider two plausible approaches:
>
>    - One may be able to prove some stability property like Hölder continuity $|b_r^n - b_s^n| \le |b_r^0 - b_s^0|^\alpha$ of the $n$-th gradient descent step. This would provide some control of the grid size $h_r$ during gradient descent iteration, given the grid size at the initial. With some excess resolution at the initial, to counter balance deteriorating stability for large $n$, this may provide the resolution condition throughout gradient descent training.
>
>    - It is plausible that the critical points satisfy the resolution condition, even if the initial values do not. Indeed, both the resolution condition and the optimal placement of the $b_r$ require fine grid size $h_r$ in regions where the target $f$ is complicated. This may allow a self improving pattern along the following lines: If the initial weights lack resolution, but allow gradient descent to roughly identify an area where $f$ is complicated, it should move resources to this area. This leads to better resolution and in turn allows gradient descent to better see where to move resources. Moving resources accordingly allows better resolution, and so on ... Such an analysis likely requires a careful multi-level analysis and is expected to be difficult even for the simple network in the manuscript.
>
>    We've added a discussion of the first item to Section 5.3.

---

### Decision · Action_Editor_bE4f · 2024-09-27

**Recommendation:** Accept with minor revision

**Comment:**

Overall, this work examines the network’s nonlinearity, moving beyond the oversimplifications of over-parameterization. The findings provide valuable insights into the complex optimization landscape of real-world neural networks. All reviewers acknowledge the technical contributions of this paper and recommend acceptance.

Very minor comment: fix the reference issues at "f a PDE ???." (right after eq. (2)), and at "capture sub-grid oscillations ???."  (at the end of  Section 5.1).

**Audience:**

Could be of interest to TMLR's audience working on the optimization of deep neural networks

**Claims And Evidence:**

This paper investigates the properties of critical points in training neural networks, focusing on a simple shallow network in one dimension. These can be addressed by ensuring sufficiently high local resolution. The results show that for “good” critical points, approximation error scales optimally with the number of parameters, while "bad” critical points may have an arbitrarily large error that is not reducible by gradient-based technique alone.